# A Biased Graph Neural Network Sampler with Near-Optimal Regret

**Qingru Zhang**[1]*     **David Wipf**[2]     **Quan Gan**[2]     **Le Song**[1,3]

[1]Georgia Institute of Technology    [2]Amazon Shanghai AI Lab
[3]Mohamed bin Zayed University of Artificial Intelligence
qingru.zhang@gatech.edu, daviwipf@amazon.com
quagan@amazon.com, le.song@mbzuai.ac.ae

## Abstract

Graph neural networks (GNN) have recently emerged as a vehicle for applying deep network architectures to graph and relational data. However, given the increasing size of industrial datasets, in many practical situations the message passing computations required for sharing information across GNN layers are no longer scalable. Although various sampling methods have been introduced to approximate full-graph training within a tractable budget, there remain unresolved complications such as high variances and limited theoretical guarantees. To address these issues, we build upon existing work and treat GNN neighbor sampling as a multi-armed bandit problem but with a newly-designed reward function that introduces some degree of bias designed to reduce variance and avoid unstable, possibly-unbounded pay outs. And unlike prior bandit-GNN use cases, the resulting policy leads to near-optimal regret while accounting for the GNN training dynamics introduced by SGD. From a practical standpoint, this translates into lower variance estimates and competitive or superior test accuracy across several benchmarks.

## 1 Introduction

Graph convolution networks (GCN) and Graph neural networks (GNN) in general [21, 17] have recently become a powerful tool for representation learning for graph structured data [6, 2, 33]. These neural networks iteratively update the representation of a node using a graph convolution operator or message passing operator which aggregate the embeddings of the neighbors of the node, followed by a non-linear transformation. After stacking multiple graph convolution layers, these models can learn node representations which can capture information from both immediate and distant neighbors.

GCNs and variants [32] have demonstrated the start-of-art performance in a diverse range of graph learning prolems [21, 17, 3, 30, 13, 15, 23]. However, they face significant computational challenges given the increasing sizes of modern industrial datasets. The multilayers of graph convolutions is equivalent to recursively unfold the neighbor aggregation in a top-down manner which will lead to an exponentially growing neighborhood size with respect to the number of layers. If the graph is dense and scale-free, the computation of embeddings will involve a large portion of the graph even with a few layers, which is intractable for large-scale graph [21, 34].

Several sampling methods have been proposed to alleviate the exponentially growing neighborhood sizes, including node-wise sampling [17, 9, 24], layer-wise sampling [8, 37, 20] and subgraph sampling [10, 35, 19]. However, the optimal sampler with minimum variance is a function of the neighbors' embeddings unknown apriori before the sampling and only partially observable for those sampled neighbors. Most previous methods approximate the optimal sampler with a static distribution which cannot reduce variance properly. And most of existing approaches [8, 37, 20, 10, 35, 19]

---

*Corresponding author. Work done during the internship at Amazon Shanghai AI Lab.

do not provide any asymptotic convergence guarantee on the sampling variance. We are therefore less likely to be confident of their behavior as GNN models are applied to larger and larger graphs. Recently, Liu et al. [24] propose a novel formulation of neighbor sampling as a multi-armed bandit problem (MAB) and apply bandit algorithms to update sampler and reduce variance. Theoretically, they provide an asymptotic regret analysis on sampling variance. Empirically, this dynamic sampler named as BanditSampler is more flexible to capture the underlying dynamics of embeddings and exhibits promising performance in a variety of datasets.

However, we will show in Section 2.3 that there are several critical issues related to the numerical stability and theoretical limitations of the BanditSampler [24]. First, the reward function designed is numerically unstable. Second, the bounded regret still can be regarded as a linear function of training horizon $T$. Third, their analysis relies on two strong implicit assumptions, and does *not* account for the unavoidable dependency between embedding-dependent rewards and GNN training dynamics.

In this paper, we build upon the bandit formulation for GNN sampling and propose a newly-designed reward function that trades bias with variance. In Section 3.1, we highlight that the proposed reward has the following crucial advantages: (i) It is numerically stable. (ii) It leads to a more meaningful notion of regret directly connected to *sampling approximation error*, the expected error between aggregation from sampling and that from full neighborhood. (iii) Its variation can be formulated by GNN training dynamics. Then in Section 3.2, we clarify how the induced regret is connected to sampling approximation error and emphasize that the bounded variation of rewards is essential to derive a meaningful sublinear regret, i.e., a per-iteration regret that decays to zero as $T$ becomes large. In that sense, we are the first to explicitly account for GNN training dynamic due to stochastic gradient descent (SGD) so as to establish a bounded variation of embedding-dependent rewards, which we present in Section 3.3.

Based on that, in Section 4, we prove our main result, namely, that the regret of the proposed algorithm as the order of $(T\sqrt{\ln T})^{2/3}$, which is near-optimal and manifest that the sampling approximation error of our algorithm asymptotically converges to that of the optimal oracle with the near-fastest rate. Hence we name our algorithm as *Thanos* from "Thanos Has A Near-Optimal Sampler". Finally, empirical results in Section 5 demonstrate the improvement of Thanos over BanditSampler and others in terms of variance reduction and generalization performance.

## 2 Background

### 2.1 Graph Neural Networks and Neighbor Sampling

**Graph Neural Networks**. Given a graph $\mathcal{G} = (\mathcal{V}, \mathcal{E})$, where $\mathcal{V}$ and $\mathcal{E}$ are node and edge sets respectively, the forward propagation of a GNN is formulated as $\boldsymbol{h}_{v,t}^{(l+1)} = \sigma(\sum_{i \in \mathcal{N}_v} a_{vi} \boldsymbol{h}_{i,t}^{(l)} W_t^{(l)})$ for the node $v \in \mathcal{V}$ at training iteration $t$. Here $\boldsymbol{h}_{i,t}^{(l)} \in \mathbb{R}^d$ is the hidden embedding of node $i$ at the layer $l$, $\boldsymbol{h}_{i,t}^{(0)} = \boldsymbol{x}_i$ is the node feature, and $\sigma(\cdot)$ is the activation function. Additionally, $\mathcal{N}_v$ is the neighbor set of node $v$, $D_v = |\mathcal{N}_v|$ is the degree of node $v$, and $a_{vi} > 0$ is the edge weight between node $v$ and $i$. And $W_t^{(l)} \in \mathbb{R}^{d \times d}$ is the GNN weight matrix, learned by minimizing the stochastic loss $\widehat{\mathcal{L}}$ with SGD. Finally, we denote $\boldsymbol{z}_{i,t}^{(l)} = a_{vi} \boldsymbol{h}_{i,t}^{(l)}$ as the weighted embedding, $[D_v] = \{i | 1 \le i \le D_v\}$, and for a vector $\boldsymbol{x} \in \mathbb{R}^{d_0}$, we refer to its 2-norm as $\|\boldsymbol{x}\|$; for matrix $W$, its spectral norm is $\|W\|$.

**Neighbor Sampling**. Recursive neighborhood expansion will cover a large portion of the graph if the graph is dense or scale-free even within a few layers. Therefore, we consider to neighbor sampling methods which samples $k$ neighbors under the distribution $p_{v,t}^{(l)}$ to approximate $\sum_{i \in \mathcal{N}_v} \boldsymbol{z}_{i,t}^{(l)}$ with this subset $\mathcal{S}_t$. We also call $p_{v,t}^{(l)}$ the policy. For ease of notation, we simplify $p_{v,t}^{(l)}$ as $p_t = \{p_{i,t} | i \in \mathcal{N}_v\}$; $p_{i,t}$ is the probability of neighbor $i$ to be sampled. We can then approximate $\boldsymbol{\mu}_{v,t}^{(l)} = \sum_{i \in \mathcal{N}_v} \boldsymbol{z}_{i,t}^{(l)}$ with an unbiased estimator $\widehat{\boldsymbol{\mu}}_{v,t}^{(l)} = \frac{1}{k} \sum_{i \in \mathcal{S}_t} \boldsymbol{z}_{i,t}^{(l)} / p_{i,t}$. As it is unbiased, only the variance term $\mathbf{V}_{p_t}(\widehat{\boldsymbol{\mu}}_{v,t}^{(l)})$ need to be considered when optimizing the policy $p_t$. Define the variance term when $k = 1$ as $\mathbf{V}_{p_t}$. Then following [29], $\mathbf{V}_{p_t}(\widehat{\boldsymbol{\mu}}_{v,t}^{(l)}) = \mathbf{V}_{p_t}/k$ with $\mathbf{V}_{p_t}$ decomposes as $\mathbf{V}_{p_t} = \mathbf{V}_e - \mathbf{V}_c$. with $\mathbf{V}_e = \sum_{i \in \mathcal{N}_v} \|\boldsymbol{z}_{i,t}^{(l)}\|^2 / p_{i,t}$, which is dependent on $p_t$ and thus refereed as the *effective variance*. And $\mathbf{V}_c = \|\sum_{j \in \mathcal{N}_v} \boldsymbol{z}_{j,t}^{(l)}\|^2$ is independent on the policy and therefore referred to as *constant variance*.

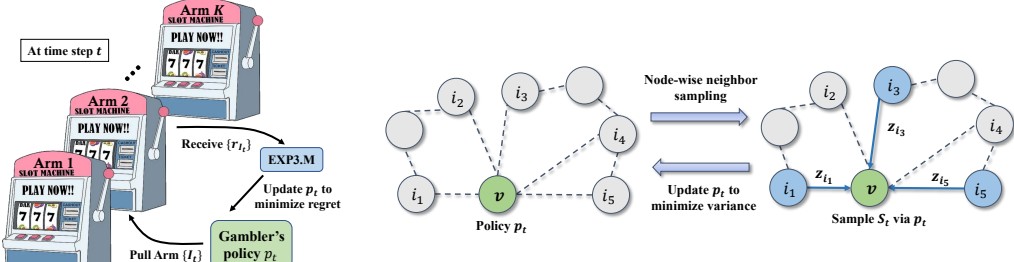

(a) Adversary Multi-Armed Bandit.   (b) Formulate neighbor sampling as a MAB problem.

Figure 1: Fig. 1a visualizes the pipeline of adversary multi-armed bandit, in which, the reward is prior unknown, non-stationary and only partially observable for the pulled arms. It motivates us to formulate the neighbor sampling as a MAB problem (Fig. 1b).

## 2.2 Formulate Neighbor Sampling as Multi-Armed Bandit

The optimal policy in terms of reducing the variance $\mathbf{V}_{p_t}$ is given by $p^*_{i,t} = \frac{\|z_{i,t}\|}{\sum_{j \in \mathcal{N}_v} \|z_{j,t}\|}$ [29]. However, this expression is intractable to compute for the following reasons: (i) It is only after sampling and forward propagation that we can observe $z^{(l)}_{i,t}$, and $z^{(l)}_{i,t}$ changes with time along an optimization trajectory with unknown dynamics. (ii) $z^{(l)}_{i,t}$ is only partially observable in that we cannot see the embeddings of the nodes we do not sample. While static policies [17, 8, 37] are capable of dealing with (ii), they are not equipped to handle (i) as required to approximate $p^*_t$ and reduce the sampling variance effectively. In contrast, adversarial MAB frameworks are capable of addressing environments with unknown, non-stationary dynamics and partial observations alike (See Fig.1). The basic idea is that a hypothetical gambler must choose which of $K$ slot machines to play (See Fig. 1a). For neighbor sampling, $K$ is equal to the degree $D_v$ of root node $v$. At each time step, the gambler takes an action, meaning pulling an arm $I_t \in [K]$ according to his policy $p_t$, and then receives a reward $r_{I_t}$. To maximize cumulative rewards, an algorithm is applied to update the policy based on the observed reward history $\{r_{I_\tau} : \tau = 1 \ldots, t\}$.

Liu et al. [24] formulate node-wise neighbor sampling as a MAB problem. Following the general strategy from Salehi et al. [29] designed to reduce the variance of stochastic gradient descent, they apply an adversarial MAB to GNN neighbor sampling using the reward

$$r_{i,t} = -\nabla_{p_{i,t}} \mathbf{V}_e(p_t) = \|z^{(l)}_{i,t}\| / p^2_{i,t}, \tag{1}$$

which is the negative gradient of the effective variance w.r.t. the policy. Since $\mathbf{V}_e(p_t) - \mathbf{V}_e(p^*_t) \leq \langle p_t - p^*_t, \nabla_{p_t} \mathbf{V}_e(p_t) \rangle$, maximizing this reward over a sequence of arm pulls, i.e., $\sum_{t=1}^T r_{I_t,t}$, is more-or-less equivalent to minimizing an upper bound on $\sum_{t=1}^T \mathbf{V}_e(p_t) - \sum_{t=1}^T \mathbf{V}_e(p^*_t)$. The actual policy is then updated using one of two existing algorithms designed for adversarial bandits, namely Exp3 [1] and Exp3.M [31]. Please see Appendix C for details. Finally, Liu et al. [24] prove that the resulting BanditSampler can asymptotically approach the optimal variance with a factor of *three*:

$$\sum_{t=1}^T \mathbf{V}_e(p_t) \leq 3 \sum_{t=1}^T \mathbf{V}_e(p^*_t) + 10\sqrt{T D_v^4 \ln(D_v/k)/k^3}. \tag{2}$$

Critically however, this result relies on strong implicit assumptions, and does *not* account for the unavoidable dependency between the reward distribution and GNN model training dynamics. We elaborate on this and other weaknesses of the BanditSampler next.

## 2.3 Limitation of BanditSampler

Updated by Exp3, BanditSampler as described is sufficiently flexible to capture the embeddings' dynamics and give higher probability to ones with larger norm. And the dynamic policy endows it with promising performance on large datasets. Moreover, it can be applied not only to GCN but GAT models [32], where $a_{vi}$ change with time as well. It is an advantage over previous sampling approaches. Even so, we still found several crucial drawbacks of the BanditSampler.

**Numerical Instability** Due to the $p_{i,t}$ in the denominator of (1), the reward of BanditSampler suffers from numerical instability especially when the neighbors with small $p_{i,t}$ are sampled. From Fig. 5a (in Appendix), we can observe that the rewards (1) of BanditSampler range between a large scale. Even though the mean of received rewards is around 2.5, the max of received rewards can attain 1800. This extremely heavy tail distribution forces us to choose a quite small temperature hyperparameter $\eta$ (Algorithm 3 and 5 in Appendix C), resulting in dramatic slowdown of the policy optimization. By

contrast, the reward proposed by us in the following section is more numerically stable (See Fig. 5b in Appendix) and possesses better practical interpretation (Fig. 2c).

**Limitation of Existing Regret and Rewards** There are two types of regret analyses for bandit algorithms [1, 4]: (i) the *weak regret* with a static oracle given by $\widehat{\mathcal{R}}(T) = \max_{j \in [D_v]}(\sum_{t=1}^{T} \mathbb{E}[r_{j,t}]) - \sum_{t=1}^{T} \mathbb{E}[r_{I_t,t}]$, which measures performance relative pulling the single best arm; and (ii) the *worst-case regret* with a dynamic oracle given by $\mathcal{R}(T) = \sum_{t=1}^{T} \max_{j \in [D_v]} \mathbb{E}[r_{j,t}] - \sum_{t=1}^{T} \mathbb{E}[r_{I_t,t}]$, where the oracle can pull the best arm at each $t$. When the growth of the regret as a function of $T$ is sublinear, the policy is long-run average optimal, meaning the long-run average performance converges to that of the oracle. But from this perspective, the bound from (2) can actually function more like worst-case regret. To see this, note that the scale factor on the oracle variance is 3, which implies that once we subtract $\mathbf{V}_e(p_t^*)$ from the upper bound, the effective regret satisfies $\mathcal{R}(T) \leq 2\sum_{t=1}^{T} \mathbf{V}_e(p_t^*) + O(\sqrt{T})$. By substituting $p_t^*$ into $\mathbf{V}_e$, we obtain $\mathbf{V}_e(p_t^*) = \sum_{i \in \mathcal{N}_v} \sum_{j \in \mathcal{N}_v} \|\mathbf{z}_{i,t}^{(l)}\| \|\mathbf{z}_{j,t}^{(l)}\|$, which can be regarded as a constant lower bound given the converged variation of $\mathbf{z}_{i,t}$ (Lemma 1). Consequently, the regret is still linear about $T$. And linear worst-case regret cannot confirm the effectiveness of policy since uniform random guessing will also achieve linear regret.

**Crucial Implicit Assumptions** There are two types of adversaries: if the current reward distribution is independent with the previous actions of the player, it is an oblivious adversary; otherwise, it is a non-oblivious adversary [7]. GNN neighbor sampling is apparently non-oblivious setting but it is theoretically impossible to provide any meaningful guarantees on the worst-case regret in the non-oblivious setting (beyond what can be achieved by random guessing) unless explicit assumptions are made on reward variation [4]. BanditSampler [24] circumvents this issue by implicitly assuming bounded variation and oblivious setting (See Appendix H), but this cannot possibly be true since embedding-dependent rewards must depend on training trajectory and previous sampling. In contrast, *we are the first to explicitly account for training dynamic in deriving reward variation and further regret bound in non-oblivious setting, and without this consideration no meaningful bound can possibly exist.*

## 3 Towards a More Meaningful Notion of Regret

To address the limitations of the BanditSampler, we need a new notion of regret and the corresponding reward upon which it is based. In this section we motivate a new biased reward function, interpret the resulting regret that emerges, and then conclude by linking with the GCN training dynamics.

### 3.1 Rethinking the Reward

Consider the following bias-variance decomposition of approximation error:

$$\mathbb{E}[\|\widehat{\boldsymbol{\mu}}_{v,t}^{(l)} - \boldsymbol{\mu}_{v,t}^{(l)}\|^2] = \|\boldsymbol{\mu}_{v,t}^{(l)} - \mathbb{E}[\widehat{\boldsymbol{\mu}}_{v,t}^{(l)}]\|^2 + \mathbf{V}_{p_t}(\widehat{\boldsymbol{\mu}}_{v,t}^{(l)}) \triangleq \text{Bias}(\widehat{\boldsymbol{\mu}}_{v,t}^{(l)}) + \mathbf{V}_{p_t}(\widehat{\boldsymbol{\mu}}_{v,t}^{(l)}).$$

Prior work has emphasized the enforcement of zero bias as the starting point when constructing samplers; however, we will now argue that broader estimators that *do* introduce bias should be reconsidered for the following reasons: (i) Zero bias itself may not be especially necessary given that even an unbiased $\widehat{\boldsymbol{\mu}}_{v,t}^{(l)}$ will become biased for approximating $\boldsymbol{h}_{v,t}^{(l+1)}$ once it is passed through the non-linear activation function. (ii) BanditSampler only tackles the variance reduction after enforcing zero bias in the bias-variance trade-off. However, it is not clear that the optimal approximation error must always be achieved via a zero bias estimator, i.e., designing the reward to minimize the approximation error in aggregate could potentially perform better, even if the estimator involved is biased. (iii) Enforcing a unbiased estimator induces other additional complications: the reward can become numerically unstable and hard to bound in the case of a non-oblivious adversary. And as previously argued, meaningful theoretical analysis must account for optimization dynamics that fall under the non-oblivious setting. Consequently, to address these drawbacks, we propose to trade variance with bias by adopting the biased estimator: $\widehat{\boldsymbol{\mu}}_{v,t}^{(l)} = \frac{D_v}{k} \sum_{i \in \mathcal{S}_t} \mathbf{z}_{i,t}^{(l)}$ and redefine the reward:

$$r_{i,t} = 2\mathbf{z}_{i,t}^{(l)\top} \bar{\mathbf{z}}_{v,t}^{(l)} - \left\| \mathbf{z}_{i,t}^{(l)} \right\|^2, \quad \text{with} \quad \bar{\mathbf{z}}_{v,t}^{(l)} = \frac{1}{D_v} \boldsymbol{\mu}_{v,t}^{(l)} = \frac{1}{D_v} \sum_{j \in \mathcal{N}_v} \mathbf{z}_{j,t}^{(l)}. \tag{3}$$

Equation (3) is derived by weighting the gradient of bias and variance w.r.t. $p_{i,t}$ equally, which we delegate to Appendix. Additionally, because of partial observability, we approximate $\bar{\mathbf{z}}^{(l)}$ with $\frac{1}{k} \sum_{i \in \mathcal{S}_t} \mathbf{z}_{i,t}^{(l)}$. We also noticed, due to the exponential function from the Exp3 algorithm (see line

6, Algorithm 3), the negative rewards of some neighbors will shrink $w_{i,t}$ considerably, which can adversely diminish their sampling probability making it hard to sample these neighbors again. Consequently, to encourage the exploration on the neighbors with negative rewards, we add ReLU function over rewards (note that our theory from Section 4 will account for this change). The practical reward is then formulated as

$$\tilde{r}_{i,t} = \text{ReLU}\Big(2\boldsymbol{z}^{(l)\top}\sum_{j\in\mathcal{S}_t}\frac{1}{k}\boldsymbol{z}_{j,t}^{(l)} - \|\boldsymbol{z}_{i,t}^{(l)}\|^2\Big). \tag{4}$$

The intuition of (3) and by extension (4) is that the neighbors whose weighted embeddings $\boldsymbol{z}_{i,t}^{(l)}$ are closer to $\bar{\boldsymbol{z}}_{v,t}^{(l)}$ will be assigned larger rewards (See Fig. 2c). Namely, our reward will bias the policy towards neighbors that having contributed to the accurate approximation instead of ones with large norm as favored by BanditSampler. And in the case of large but rare weighted embeddings far from $\boldsymbol{\mu}_{v,t}^{(l)}$, BanditSampler tends to frequently sample these large and rare embeddings, causing significant deviations. The empirical evidence is shown in Section 5.3.

The reward (3) possesses following practical and theoretical advantages, which will be expanded more in next sections:

- Since it is well bounded by $r_{i,t} = \|\bar{\boldsymbol{z}}_{v,t}^{(l)}\|^2 - \|\boldsymbol{z}_{i,t}^{(l)} - \bar{\boldsymbol{z}}_{v,t}^{(l)}\|^2 \le \|\bar{\boldsymbol{z}}_{v,t}^{(l)}\|^2$, the proposed reward is more numerical stable as we show in Fig. 5 (See Appendix).
- It will incur a more meaningful notion of regret, meaning the regret defined by (3) is equivalent to the gap between the policy and the oracle w.r.t. approximation error.
- The variation of reward (3) is tractable to bound as a function of training dynamics of GCN in non-oblivious setting, leading to a provable sublinear regret as the order of $(D_v \ln D_v)^{1/3}(T\sqrt{\ln T})^{2/3}$, which means the approximation error of policy asymptotically converges to the optimal oracle with a factor of *one* rather than three.

### 3.2 Interpreting the Resulting Regret

We focus on the worst-case regret in the following analysis. The regret defined by reward (3) is directly connected to approximation error. More specifically, we notice $r_{i,t} = -\|\boldsymbol{z}_{i,t}^{(l)} - \bar{\boldsymbol{z}}_{v,t}^{(l)}\|^2 + \|\bar{\boldsymbol{z}}_{v,t}^{(l)}\|^2$. Since $\|\bar{\boldsymbol{z}}_{v,t}^{(l)}\|^2$ will be canceled out in $\mathcal{R}(T)$, we have $\mathcal{R}(T) = \sum_{t=1}^{T}(\mathbb{E}\|\boldsymbol{z}_{I_t,t}^{(l)} - \bar{\boldsymbol{z}}_{v,t}^{(l)}\|^2 - \max_{j\in[D_v]}\mathbb{E}\|\boldsymbol{z}_{j,t}^{(l)} - \bar{\boldsymbol{z}}_{v,t}^{(l)}\|^2)$, where the former term is the expected approximation error of the policy and the latter is that of the optimal oracle. Consequently, the regret defined by (3) is the gap between the policy and the optimal oracle w.r.t. the approximation error.

Then we clarify how to bound this regret. The worst-case regret is a more solid guarantee of optimality than the weak regret. Even though some policies can establish the best achievable weak regret $O(\sqrt{T})$, their worst-case regret still be linear. This is because the gap between static and dynamic oracles can be a linear function of $T$ if there is no constraint on rewards. For example, consider the following worst-case scenario. Given three arms $\{i_1, i_2, i_3\}$, at every iteration, one of them will be assigned a reward of 3 while the others receive only 1. In that case, consistently pulling any arm will match the static oracle and any static oracle will have a linear gap with the dynamic oracle. Hence it is impossible to establish a sublinear worst-case regret unless additional assumptions are introduced on the variation of the rewards to bound the gap between static and dynamic oracles [4]. Besbes et al. [4] claim that the worst-case regret can be bounded as a function of the variation budget:

$$\sum_{t=1}^{T-1}\sup_{i\in[D_v]}\left|\mathbb{E}[\tilde{r}_{i,t+1}] - \mathbb{E}[\tilde{r}_{i,t}]\right| \le V_T \tag{5}$$

where $V_T$ is called the variation budget. Then, Besbes et al. [4] derived the regret bound as $\mathcal{R}(T) = O(K\ln K \cdot V_T^{1/3}T^{2/3})$ for Rexp3. Hence, if the variation budget is a sublinear function of $T$ in the given environment, the worst-case regret will be sublinear as well.

To fix the theoretical drawbacks of BanditSampler, we first drop the assumption of oblivious adversary, i.e. considering the dependence between rewards and previous sampling along the training horizon of GCN. Then to bound the variation budget, we account for GCN training dynamic in practically-meaningful setting (i.e. no unrealistic assumptions) as described next.

### 3.3 Accounting for the Training Dynamic of GCN

One of our theoretical contributions is to study the dynamics of embeddings in the context of GNN training optimized by SGD. We present our assumptions as follows:

- **Lipschitz Continuous Activation Function:** $\forall \boldsymbol{x}, \boldsymbol{y}, \|\sigma(\boldsymbol{x}) - \sigma(\boldsymbol{y})\| \le C_\sigma \|\boldsymbol{x} - \boldsymbol{y}\|$ and $\sigma(\boldsymbol{0}) = \boldsymbol{0}$.
- **Bounded Parameters:** For any $1 \le t \le T$ and $0 \le l \le L-1$, $\|W_t^{(l)}\| \le C_\theta$.
- **Bounded Gradients:** For $1 \le t \le T$, $\exists C_g$, such that $\sum_{l=0}^{L-1} \|\nabla_{W_t^{(l)}} \widehat{\mathcal{L}}\| \le C_g$.

Besides, given the graph $\mathcal{G}$ and its feature $X$, since $a_{vi}$ is fixed in GCN, define $C_x = \max_{v \in \mathcal{V}} \|\sum_{i \in \mathcal{N}_v} a_{vi} \boldsymbol{x}_i\|$. Define $\bar{D} = \max_{v \in \mathcal{V}} D_v$, $\bar{A} = \max_{v,i} a_{vi}$, $G = C_\sigma C_\theta \bar{D}\bar{A}$, and $\Delta_{t,l}^z = \max_{i \in \mathcal{V}} \|\boldsymbol{z}_{i,t+1}^{(l)} - \boldsymbol{z}_{i,t}^{(l)}\|$. For SGD, we apply the learning rate schedule as $\alpha_t = 1/t$. The above assumptions are reasonable. The bounded gradient is generally assumed in the non-convex/convex convergence analysis of SGD [25, 28]. And the learning rate schedule is necessary for the analysis of SGD to decay its constant gradient variance [16]. Then we will bound $\Delta_{t,l}^z$ as a function of gradient norm and step size by recursively unfolding the neighbor aggregation.

**Lemma 1** (Dynamic of Embedding). *Based on our assumptions on GCN, for any $i \in \mathcal{V}$ at the layer $l$, we have:*

$$\left\| \boldsymbol{z}_{i,t}^{(l)} \right\| \le C_z, \quad \left| \tilde{r}_{i,t} \right| \le C_r, \quad \left| r_{i,t} \right| \le C_r, \tag{6}$$

*where $C_z = G^{l-1} \bar{A} C_\sigma C_\theta C_x$ and $C_r = 3C_z^2$. Then, consider the training dynamics of GCN optimized by SGD. For any node $i \in \mathcal{V}$ at the layer $l$, we have*

$$\Delta_{t,l}^z = \max_{i \in \mathcal{V}} \left\| \boldsymbol{z}_{i,t+1}^{(l)} - \boldsymbol{z}_{i,t}^{(l)} \right\| \le \alpha_t G^{l-1} \bar{A} C_\sigma C_x C_g. \tag{7}$$

Lemma 1 is obtained by recursively unfolding neighbor aggregations and training steps, and can be generally applied to any GCN in practical settings. Based on it, we can derive the variation budget of reward (3) and (4) as a function of $\Delta_{t,l}^z$ in the non-oblivious setup.

**Lemma 2** (Variation Budget). *Given the learning rate schedule of SGD as $\alpha_t = 1/t$ and our assumptions on the GCN training, for any $T \ge 2$, any $v \in \mathcal{V}$, the variation of the expected reward in (3) and (4) can be bounded as:*

$$\sum_{t=1}^{T} \left| \mathbb{E}[r_{i,t+1}] - \mathbb{E}[r_{i,t}] \right| \le V_T = \bar{C}_v \ln T, \quad \sum_{t=1}^{T} \left| \mathbb{E}[\tilde{r}_{i,t+1}] - \mathbb{E}[\tilde{r}_{i,t}] \right| \le V_T = \bar{C}_v \ln T \tag{8}$$

*where $\bar{C}_v = 12 G^{2(l-1)} C_\sigma^2 C_x^2 \bar{A}^2 C_\theta C_g$.*

The derivation of Lemma 2 is attributed to that our reward variation can be explicitly formulated as a function of embeddings' variation. In contrast, $p_{i,t}$ emerging in the denominator of (1) incurs not only the numerically unstable reward but hardship to bound its variation. More specifically, $p_{i,t}$ is proportional to the summation of observed reward history of neighbor $i$, which is hard to bound due to the complication to explicitly keep track of overall sampling trajectory as well as its bilateral dependency with $p_{i,t}$. It is potentially why BanditSampler's regret (2) ignores the dependency between rewards and previous training/sampling steps. On the contrary, our rewards are tractable to bound as a function of embeddings' dynamic in practical non-oblivious setting, leading to a sublinear variation budget (8), and further a solid near-optimal worst-case regret as presented next.

## 4   Main Result: Thanos and Near-Optimal Regret

---
**Algorithm 1** Thanos

---
1: **Input:** $\eta > 0, \gamma \in (0,1), k, T, \Delta_T, \mathcal{G}, X, \{\alpha_t\}_{t=1}^{T}$.
2: **Initialize**: For any $v \in \mathcal{V}$, any $i \in \mathcal{N}_v$, set $p_{i,1} = 1/D_v$.
3: **for** $t = 1, 2, \ldots, T$ **do**
4:    Reinitialize the policy every $\Delta_T$ steps: $\forall v \in \mathcal{V}, \forall i \in \mathcal{N}_v$, set $p_{i,t} = 1/D_v$.
5:    Sample $k$ neighbors with $p_t$ and estimate $\boldsymbol{\mu}_{v,t}^{(l)}$ with the estimator $\widehat{\boldsymbol{\mu}}_{v,t}^{(l)} = \frac{D_v}{k} \sum_{i \in \mathcal{S}_t} \boldsymbol{z}_{i,t}^{(l)}$.
6:    Forward GNN model and calculate the reward $r_{i,t}$ according to (4).
7:    Update the policy and optimize the model following [24] using $\eta$, $\gamma$, and $\{\alpha_t\}_{t=1}^{T}$.
8: **end for**

---

Algorithm 1 presents the condensed version of our proposed algorithm. See Algorithm 2 in Appendix B for the detailed version. Besides the trade-off between bias and variance, and exploration and exploitation, our proposed algorithm also accounts for a third trade-off between *remembering* and *forgetting*: given the non-stationary reward distribution, while keeping track of more observations

can decrease the variance of reward estimation, the non-stationary environment implies that "old" information is potentially less relevant due to possible changes in the underlying rewards. The changing rewards give incentive to dismiss old information, which in turn encourages exploration. Therefore, we apply Rexp3 algorithm [4] to tackle the trade-off between remembering and forgetting by reinitializing the policy every $\Delta_T$ steps (line 4 in Algorithm 1).

Then, we present our main result: bounding the worst-case regret of the proposed algorithm:

$$\mathcal{R}(T) = \sum_{t=1}^{T} \sum_{i \in \mathcal{N}_k^*} \mathbb{E}[r_{i,t}] - \sum_{t=1}^{T} \sum_{I_t \in \mathcal{S}_t} \mathbb{E}^{\pi}[r_{I_t,t}]. \tag{9}$$

where $\mathcal{N}_k^* = \mathrm{argmax}_{\mathcal{N}_k \subset \mathcal{N}_v} \sum_{i \in \mathcal{N}_k} \mathbb{E}[r_{i,t}], |\mathcal{N}_k^*| = k$. Because we consider the non-oblivious adversary, $\mathbb{E}[r_{i,t}]$ is taken over the randomness of rewards caused by the previous history of randomized arm pulling. $\mathbb{E}^{\pi}[r_{I_t,t}]$ is taken over the joint distribution $\pi$ of the action sequence $(\mathcal{S}_1, \mathcal{S}_2, \ldots, \mathcal{S}_T)$.

**Theorem 3** (Regret Bound). *Consider Algorithm 1 as the neighbor sampling algorithm for training GCN. Given either* (3) *or* (4) *as reward function, we can bound its regret as follows. Let* $\Delta_T = (\bar{C}_v \ln T)^{-\frac{2}{3}} (D_v \ln D_v)^{\frac{1}{3}} T^{\frac{2}{3}}$, $\eta = \sqrt{\frac{2k \ln(D_v/k)}{C_r(\exp(C_r)-1)D_v T}}$, *and* $\gamma = \min\{1, \sqrt{\frac{(\exp(C_r)-1)D_v \ln(D_v/k)}{2kC_r T}}\}$. *Given the variation budget in* (8), *for every* $T \geq D_v \geq 2$, *we have the regret bound for either* (3) *or* (4) *as*

$$\mathcal{R}(T) \leq \bar{C}(D_v \ln D_v)^{\frac{1}{3}} \cdot (T\sqrt{\ln T})^{\frac{2}{3}}. \tag{10}$$

*where* $\bar{C}$ *is a absolute constant independent with* $D_v$ *and* $T$.

The obtained regret is as the order of $(D_v \ln D_v)^{1/3}(T\sqrt{\ln T})^{2/3}$. According to Theorem 1 in [4], the worst-case regret of any policy is lower bounded by $O((D_v V_T)^{1/3}T^{2/3})$, suggesting our algorithm is near-optimal (with a modest $\ln T$ factor from optimal). In that sense, we name our algorithm as *Thanos* from "Thanos Has A Near-Optimal Sampler."

The near-optimal regret from Theorem 3 can be obtained due to the following reasons: (i) Our proposed reward leads to a more meaningful notion of regret which is directly connected to approximation error. (ii) Its variation budget is tractable to be formulated by the dynamic of embeddings. (iii) We explicitly study training dynamic of GCN to bound embeddings' dynamic by recursively unfolding the neighbor aggregation and training steps in the practical setting.

As mentioned in Section 3.2, the regret based on rewards (3) is equivalent to approximation error. The result of Theorem 3 says the approximation error of Thanos asymptotically converges to that of the optimal oracle with the near-fastest convergence rate. In the case of enforcing zero bias like BanditSampler, sampling variance is the exact approximation error. However, even if we ignore other previously-mentioned limitations, its regret (2) suggests the approximation error of their policy asymptotically converges to three (as opposed to one) times of the oracle's approximation error, so the regret is still linear. We compare the existing theoretical convergence guarantees in Table 1.

Table 1: Comparison of existing asymptotic convergence guarantees.

| | Dyanmic policy | Convergence analysis | Theory accounts for practical training | Bound reward variation explicitly | Sublinear gap to the optimal oracle | Stable reward/policy |
|---|---|---|---|---|---|---|
| Uniform policy | ✗ | ✗ | ✗ | ✗ | ✗ | ✓ |
| BanditSampler | ✓ | ✓ | ✗ | ✗ | ✗ | ✗ |
| Thanos | ✓ | ✓ | ✓ | ✓ | ✓ | ✓ |

# 5 Experiments

We describe the experiments to verify the effectiveness of Thanos and its improvement over Bandit-Sampler in term of sampling approximation error and final practical performance.

## 5.1 Illustrating Policy Differences via Synthetic Stochastic Block Model Data

As mentioned in Section 3.1, our reward will bias to sample the neighbors having contributed to accurate approximation. Fig. 2c is the visualization of this intuition: after setting $\bar{z}_{v,t} = (1,1)^{\top}$,

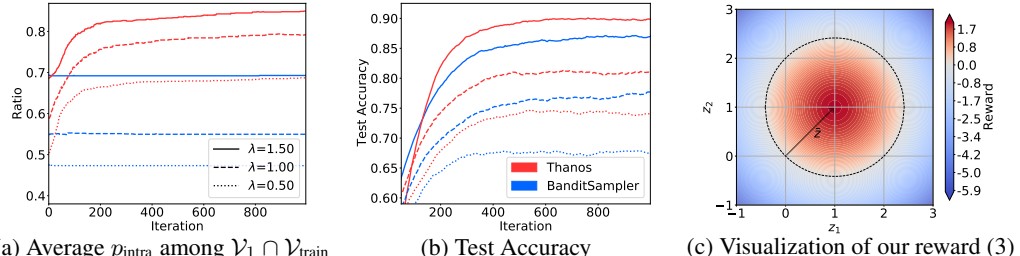

(a) Average $p_{\text{intra}}$ among $\mathcal{V}_1 \cap \mathcal{V}_{\text{train}}$  (b) Test Accuracy  (c) Visualization of our reward (3)

Figure 2: Figs. 2a and 2b illustrate the policy difference and compare their practical performance via cSBM synthetic data. Fig. 2c plot our reward function (3) after setting $\bar{\boldsymbol{z}}_{v,t}^{(l)} = (1,1)^\top$.

the reward inside the dashed circle is positive; otherwise negative. And the embeddings closer to $\bar{\boldsymbol{z}}_{v,t}$ will have larger rewards. In order to understand how this bias differentiates the policy of two samplers given different distribution of features and edges, we propose to use cSBM[14, 11] to generate synthetic graphs. We consider a cSBM [14] with two classes, whose node set $\mathcal{V}_1$ and $\mathcal{V}_2$ have 500 nodes. The node features are sampled from class-specific Gaussians $N_1, N_2$. We set feature size to 100, average degree $2\bar{d} = 20$, $k = 10$, and $\mu = 1$, and we note that $\mu$ controls the difference between two Gaussian's mean [14]. The average number of inter-class and intra-class edges per node is $\bar{d} - \lambda\bar{d}^{1/2}$ and $\bar{d} + \lambda\bar{d}^{1/2}$ respectively. Then, we scale down the node features of $\mathcal{V}_1$ by 0.1 to differentiate the distribution of feature norm and test the sampler's sensitivity to it. The configuration of training and samplers is same as Section 5.4 and listed in Appendix.

In the case of cSBMs, an ideal sampler should sample more intra-class neighbors than inter-class neighbors to get linear-separable embeddings and better classification. Thus, we inspect for each $v$ the $k$ neighbors having the top-$k$ highest sampling probability, and compute the ratio of intra-class neighbors among them, i.e. $p_{\text{intra}} = \sum_{i \in \mathcal{N}_v} \mathbf{1}\{(y_i = y_v) \cap (p_{i,t} \text{ is top-}k)\}/k$. We report the average of $p_{\text{intra}}$ for $\mathcal{V}_1 \cap \mathcal{V}_{\text{train}}$ versus $t$ in Fig. 2a. For the scaled community $\mathcal{V}_1$, Thanos will be biased to sample more intra-class neighbors due to the intuition explained by Fig. 2c, leading to more accurate approximation and improvement on test accuracy over BanditSampler as shown in Fig. 3a and 2b. This claim holds true under different edge distributions ($\lambda \in \{0.5, 1, 1.5\}$). We additionally report the results on unscaled $\mathcal{V}_2$ for comparison in Appendix.

## 5.2 Evaluating the Sampling Approximation Error

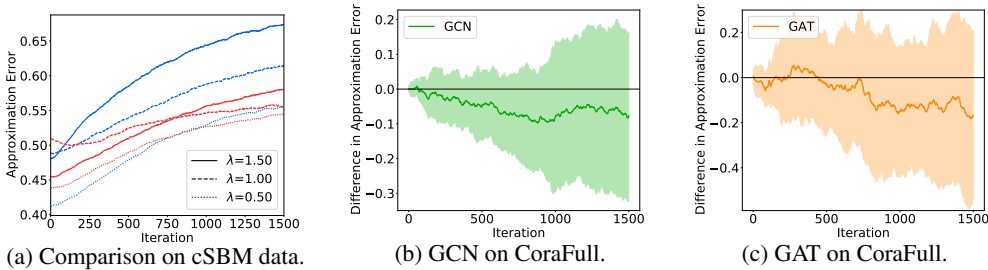

(a) Comparison on cSBM data.  (b) GCN on CoraFull.  (c) GAT on CoraFull.

Figure 3: Compare the approximation error between two samplers in cSBM synthetic graphs (Fig. 3a) and training of GCN (Fig. 3b) and GAT (Fig. 3c) on Cora. In Fig. 3b and 3c, negative values indicate that Thanos has a lower approximation error than BanditSampler.

We numerically compare the approximation error between two samplers in the training of GCN and GAT on Cora dataset from Kipf and Welling [21] as well as cSBM synthetic data in Section 5.1. At each iteration, given a batch of nodes $\mathcal{V}_L$ at the top layer, we perform sampling with BanditSampler and Thanos respectively, getting two subgraphs $\mathcal{G}_{bs}$ and $\mathcal{G}_{our}$. For Cora, we perform forward propagation on the original graph $\mathcal{G}$ as well as $\mathcal{G}_{bs}$ and $\mathcal{G}_{our}$ respectively with the same model parameters $\{W_t^{(l)}\}_{l \in [L]}$, and we get the accurate $\boldsymbol{\mu}_{v,t}^{(1)}$ of the first layer aggregation as well as its estimated values $\widehat{\boldsymbol{\mu}}_{v,t}^{(bs)}$ and $\widehat{\boldsymbol{\mu}}_{v,t}^{(our)}$ from both samplers. We compute $\text{dist}_{bs} = \sum_{v \in \mathcal{V}_L} \|\widehat{\boldsymbol{\mu}}_{v,t}^{(bs)} - \boldsymbol{\mu}_{v,t}^{(1)}\|$ and $\text{dist}_{our} = \sum_{v \in \mathcal{V}_L} \|\widehat{\boldsymbol{\mu}}_{v,t}^{(our)} - \boldsymbol{\mu}_{v,t}^{(1)}\|$. We set $k = 2$, $\gamma = 0.1$, $\eta = 0.1$ for Thanos, $\eta = 0.01$ for BanditSampler (since its unstable rewards require smaller $\eta$), $\Delta_T = 200$, $\alpha_t = 0.001$, $L = 2$ and the dimension of hidden embeddings $d = 16$. Fig. 3 plots the mean and the standard deviation of $\Delta_{\text{dist}} = \sum_{t=1}^T \text{dist}_{our} - \sum_{t=1}^T \text{dist}_{bs}$ with 10 trials. The mean curves of both GCN and GAT are

below zero, suggesting Thanos establishes lower approximation error in practice. For cSBM synthetic graphs, we follow the setting as Section 5.1, compare two samplers under different edge distributions ($\lambda \in \{0.5, 1, 1.5\}$) and directly plot $\sum_t \text{dist}_{bs}$ (blue) and $\sum_t \text{dist}_{our}$ (red). From Fig. 3a, we know Thanos achieves quite lower approximation error and higher test accuracy (Fig. 2b) in the setting of less inter-edges (e.g. $\lambda = 1$ or $1.5$) due to the intuition manifested by Fig. 2c, whereas BanditSampler is biased to sample large-norm neighbors, resulting in high approximation error and degenerated performance. For small $\lambda = 0.5$, the almost-equal number of inter/intra edges will shift $\boldsymbol{\mu}_{v,t}$ to the unscaled community $\mathcal{V}_2$. Hence two samplers' approximation error are close.

## 5.3 Sensitivity to Embedding Norms

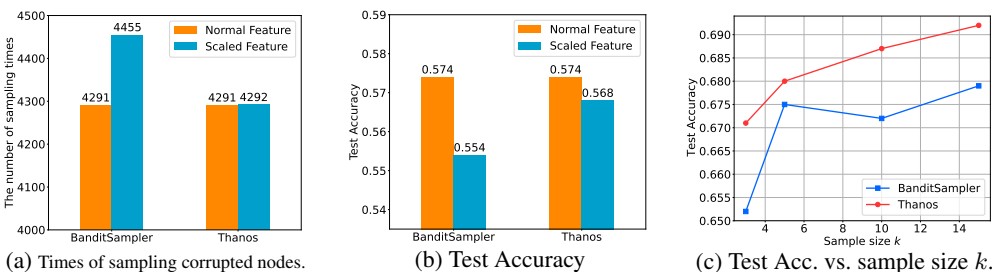

(a) Times of sampling corrupted nodes. (b) Test Accuracy (c) Test Acc. vs. sample size $k$.

Figure 4: Fig. 4a plots the average number of times the corrupted/rescaled nodes were sampled per epoch by both samplers on corrupted CoraFull. And Fig. 4b compares their corresponding test accuracy, suggesting the performance of BanditSampler will be degenerate by its sensitivity of embedding norm. Fig. 4c shows the ablation study on sample size $k$ with ogbn-arxiv.

Previously, we claim that BanditSampler will bias policy to the neighbors with large norm, potentially leading to severe deviation from $\boldsymbol{\mu}_{v,t}$ as well as a drop of performance. In this section, we present the evidence on CoraFull [5] with corrupted features and demonstrate that our algorithm resolves this issue. For CoraFull, we randomly corrupt 800 (roughly 5% of) training nodes by multiplying their features by 40. We run both samplers 300 epochs with the corrupted CoraFull and count the total number of times that these corrupted nodes were sampled per epoch. We set $k = 3, \eta = 1, \gamma = 0.2$ and the other hyperparameters the same as Section 5.4. We repeat 5 trials for each algorithm and report the average over epochs and trials. We also record the test accuracy with the best validation accuracy in each trial and report its mean across trials. From Fig. 4a, we can tell BanditSampler biases to corrupted nodes, degenerating its performance more as shown in Fig. 4b.

## 5.4 Accuracy Comparisons across Real-World Benchmark Datasets

We conduct node classification experiments on several benchmark datasets with large graphs: ogbn-arxiv, ogbn-products [18], CoraFull, Chameleon [11] and Squirrel [27]. The models include GCN and GAT. For GCN, we compare the test accuracy among Thanos, BanditSampler, GraphSage[17], LADIES[37], GraphSaint[35], ClusterGCN[10] and vanilla GCN. For GAT, we compare test accuracy among Thanos, BanditSampler and vanilla GAT. The experimental setting is similar with Liu et al. [24]. The dimension of hidden embedding $d$ is 16 for Chameleon and Squirrel, 256 for the others. The number of layer is fixed as 2. We set $k = 3$ for CoraFull; $k = 5$ for ogbn-arxiv, Chameleon, Squirrel; $k = 10$ for ogbn-products. We searched the learning rate among $\{0.001, 0.002, 0.005, 0.01\}$ and found 0.001 optimal. And we set the penalty weight of $l_2$ regularization 0.0005 and dropout rate 0.1. We do grid search for sampling hyperparameters: $\eta, \gamma,$ and $\Delta_T$ and choose optimal ones for each. Their detailed settings and dataset split are listed in Appendix. Also we apply neighbor sampling for test nodes for all methods, which is consistent with prior LADIES and GraphSaint experiments, and is standard for scalability in practical setting. From Table 2, we can tell our algorithm achieves superior performance over BanditSampler for training GAT, and competitive or superior performance for training GCN.

## 5.5 Sample Size Ablation

To verify the sensitivity of Thanos w.r.t. sample size $k$, we compare the test accuracy between Thanos and BanditSampler as sample size $k$ increases on Ogbn-arxiv. The other hyperparameter setting is the same as Section 5.4. We compare two samplers with $k = 3, k = 5, k = 10, k = 15$. The result from Fig. 4c suggests Thanos still exhibits a mainfest improvement over BanditSampler as $k$ increases.

Table 2: Test accuracy. '×' means the program crashed after a few epochs due to the massive memory cost and segmentation fault in TensorFlow. Bold indicates first; red second.

|  | Methods | Test Accuracy | | | | |
|---|---|---|---|---|---|---|
|  |  | Chameleon | Squirrel | Ogbn-arxiv | CoraFull | Ogbn-products |
| GCN | Vanilla GCN | 0.518(±0.021) | 0.327(±0.023) | 0.659(±0.004) | 0.565(±0.004) | × |
|  | GraphSage | 0.559(±0.013) | 0.385(±0.007) | 0.652(±0.005) | 0.554(±0.004) | 0.753(±0.002) |
|  | LADIES | 0.547(±0.008) | 0.338(±0.021) | 0.651(±0.003) | 0.564(±0.001) | 0.673(±0.004) |
|  | GraphSaint | 0.525(±0.022) | 0.352(±0.007) | 0.565(±0.002) | **0.583**(±0.003) | 0.746(±0.005) |
|  | ClusterGCN | 0.577(±0.022) | 0.391(±0.015) | 0.575(±0.004) | 0.390(±0.005) | 0.746(±0.014) |
|  | BanditSampler | 0.578(±0.016) | 0.383(±0.005) | 0.652(±0.005) | 0.555(±0.009) | 0.754(±0.007) |
|  | Thanos | **0.607**(±0.012) | **0.401**(±0.013) | **0.663**(±0.006) | 0.574(±0.010) | **0.759**(±0.001) |
| GAT | Vanilla GAT | 0.558(±0.009) | 0.339(±0.011) | 0.682(±0.005) | 0.519(±0.012) | × |
|  | BanditSampler | 0.602(±0.005) | 0.386(±0.006) | 0.675(±0.002) | 0.544(±0.002) | 0.756(±0.001) |
|  | Thanos | **0.620**(±0.014) | **0.412**(±0.003) | **0.680**(±0.001) | **0.559**(±0.011) | **0.759**(±0.002) |

## 6 Related Work

Hamilton et al. [17] initially proposed to uniformly sample subset for each root node. Many other methods extend this strategy, either by reducing variance [9], by redefining neighborhoods [34] [36] [22], or by reweighting the policy with MAB [24] and reinforcement learning [26]. Layer-wise sampling further reduces the memory footprint by sampling a fixed number of nodes for each layer. Recent layer-wise sampling approaches include [8] and [37] that use importance sampling according to graph topology, as well as [20] and [12] that also consider node features. Moreover, training GNNs with subgraph sampling involves taking random subgraphs from the original graph and apply them for each step. Chiang et al. [10] partitions the original graph into smaller subgraphs before training. Zeng et al. [35] and Hu et al. [19] samples subgraphs in an online fashion. However, most of them do not provide any convergence guarantee on the sampling variance. We are therefore less likely to be confident of their behavior as GNN models are applied to larger and larger graphs.

## 7 Conclusion

In this paper, we build upon bandit formulation for GNN sampling and propose a newly-designed reward function that introduce some degree of bias to reduce variance and avoid numerical instability. Then, we study the dynamic of embeddings introduced by SGD so that bounding the variation of our rewards. Based on that, we prove our algorithm incurs a new-optimal regret. Besides, our algorithm named Thanos addresses another trade-off between remembering and forgetting caused by the non-stationary rewards by employing Rexp3 algorithm. The empirical results demonstrate the improvement of Thanos over BanditSampler in term of approximation error and test accuracy.

## Acknowledgements

We would like to thank Amazon Web Service for supporting the computational resources, Hanjun Dai for the extremely helpful discussion, and the anonymous reviewers for providing constructive feedback on our manuscript.

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
