# A    Appendix

# B    A Detailed Version of Our Algorithm

---
**Algorithm 2** Thanos

---
1: **Input:** $\eta > 0, \gamma \in (0,1), k, T, \Delta_T, \mathcal{G}, X, \{\alpha_t\}_{t=1}^T$.
2: **Initialize**: For any $v \in \mathcal{V}$, any $i \in \mathcal{N}_v$, set $w_{i,1}^{(v)} = 1$.
3: **for** $t = 1, 2, \ldots, T$ **do**
4:    **if** $t \mod \Delta_T = 0$ **then**
5:       **Reinitialize the policy:** (Rexp3) for any $v \in \mathcal{V}$, for any $i \in \mathcal{N}_v$, set $w_{i,t}^{(v)} = 1$.
6:    **end if**
7:    **for** $v \in \mathcal{V}, i \in \mathcal{N}_v$ **do**
8:       Set the policy $p_t$ with Exp3.M algorithm (See Algorithm 5).
9:    **end for**
10:   Read a batch of labeled nodes $\mathcal{V}_L$ at the top layer $L$.
11:   **for** $l = L, L-1, \ldots, 1$ **do**
12:      For any $v \in \mathcal{V}_l$, sample $k$ neighbors with DepRound algorithm: $\mathcal{S}_t = \text{DepRound}(k, p_t)$.
13:      $\mathcal{V}_{l-1} := \mathcal{V}_{l-1} + \mathcal{S}_t$.
14:   **end for**
15:   Forward GNN model and estimate $\boldsymbol{\mu}_{v,t}^{(l)}$ with the estimator $\widehat{\boldsymbol{\mu}}_{v,t}^{(l)} = \frac{D_v}{k} \sum_{i \in \mathcal{S}_t} \boldsymbol{z}_{i,t}^{(l)}$.
16:   **for** $v \in \mathcal{V}_1$ **do**
17:      For any $I_t \in \mathcal{S}_t$, calculate the reward $r_{I_t,t}$.
18:      Update the policy with Exp3 algorithm: for any $i \in \mathcal{N}_v$

$$\widehat{r}_{i,t} = \frac{r_{i,t}}{p_{i,t}} \mathbf{1}_{i \in \mathcal{S}_t}, \quad w_{i,t+1}^{(v)} = w_{i,t}^{(v)} \exp(\eta \widehat{r}_{i,t}).$$

19:   **end for**
20:   Optimize the parameters with SGD: $W_{t+1}^{(l)} = W_t^{(l)} - \alpha_t \nabla_{W_t^{(l)}} \widehat{\mathcal{L}}, 0 \le l \le L-1$.
21: **end for**

---

Same as BanditSampler, we also use the embeddings of 1-st layer to calculate rewards and update the policy, i.e. the policy of 1-st layer also serves other layers.

# C    Related Algorithms

---
**Algorithm 3** Exp3

---
1: **Input:** $\eta > 0, \gamma \in (0,1], k, T$.
2: For any $i \in [D_v]$, set $w_{i,1} = 1$.
3: **for** $t = 1, 2, \ldots, T$ **do**
4:   Compute the policy $p_t$: $\forall i \in [D_v], p_{i,t} = (1 - \gamma) \frac{w_{i,t}}{\sum_{j \in [D_v]} w_{j,t}} + \frac{\gamma}{D_v}$.
5:   Draw an arm $I_t$ according to the distribution $p_t$ and receive a reward $r_{I_t,t}$.
6:   For $i \in [D_v]$, compute $\widehat{r}_{i,t} = \frac{r_{i,t}}{p_{i,t}} \mathbf{1}_{i \in \mathcal{S}_t}, w_{i,t+1} = w_{i,t} \exp(\eta \widehat{r}_{i,t})$.
7: **end for**

---

---
**Algorithm 4** DepRound

---
1: **Input:** Sample size $k(k \le D_v)$, sample distribution $(p_1, p_2, \ldots, p_{D_v})$ with $\sum_{i=1}^{D_v} = k$.
2: **while** there is an $i$ with $0 < p_i < 1$ **do**
3:   Choose distinct $i$ and $j$ with $0 < p_i < 1$ and $0 < p_j < 1$.
4:   Set $\beta = \min\{1 - p_i, p_j\}$ and $\zeta = \min\{p_i, 1 - p_j\}$.
5:   Update $p_i$ and $p_j$ as:

$$(p_i, p_j) = \begin{cases} (p_i + \beta, p_j - \beta), & \text{with probability } \frac{\zeta}{\beta + \zeta}; \\ (p_i - \zeta, p_j + \zeta), & \text{with probability } \frac{\beta}{\beta + \zeta}. \end{cases}$$

6: **end while**
7: **return:** $\{i : p_i = 1, 1 \le i \le D_v\}$.

---

**Algorithm 5** Exp3.M

---

1: **Input:** $\eta > 0, \gamma \in (0,1], k, T$.
2: For any $i \in [D_v]$, set $w_{i,1} = 1$.
3: **for** $t = 1, 2, \ldots, T$ **do**
4:    **if** $\operatorname{argmax}_{j \in [D_v]} w_{j,t} \geq (\frac{1}{k} - \frac{\gamma}{D_v})$ **then**
5:       Decides $\bar{a}_t$ so as to satisfy

$$\frac{\bar{a}_t}{\sum_{w_{i,t} \geq \bar{a}_t} \bar{a}_t + \sum_{w_{i,t} < \bar{a}_t} w_{i,t}} = (\frac{1}{k} - \frac{\gamma}{D_v})/(1 - \gamma)$$

6:       Set $U_t = \{i : w_{i,t} \geq \bar{a}_t\}$ and $w'_{i,t} = \bar{a}_t$ for $i \in U_t$
7:    **else**
8:       Set $U_t = \emptyset$.
9:    **end if**
10:    Compute the policy $p_t$: for $i \in [D_v]$

$$p_{i,t} = k\left((1 - \gamma)\frac{w'_{i,t}}{\sum_{j=1}^{D_v} w'_{j,t}} + \frac{\gamma}{D_v}\right).$$

11:    Sample a subset $\mathcal{S}_t$ with $k$ elements: $\mathcal{S}_t = \text{DepRound}(k, (p_{1,t}, p_{2,t}, \ldots, p_{D_v,t}))$.
12:    Receive the rewards $r_{I_t,t}, I_t \in \mathcal{S}_t$.
13:    For $i \in [D_v]$, set

$$\widehat{r}_{i,t} = \begin{cases} r_{i,t}/p_{i,t}, & i \in \mathcal{S}_t; \\ 0, & \text{otherwise}, \end{cases} \qquad w_{i,t+1} = \begin{cases} w_{i,t}\exp(\eta\widehat{r}_{i,t}), & i \notin U_t \\ w_{i,t}, & \text{otherwise}. \end{cases}$$

14: **end for**

---

# D   The Derivation of Reward (3)

For the biased estimator:

$$\widehat{\boldsymbol{\mu}}_{v,t}^{(l)} = \frac{D_v}{k} \sum_{i \in \mathcal{S}_t} a_{vi} \boldsymbol{h}_{i,t}^{(l)}.$$

We have its bias-variance decomposition as

$$\mathbb{E}[\|\widehat{\boldsymbol{\mu}}_{v,t}^{(l)} - \boldsymbol{\mu}_{v,t}^{(l)}\|^2] = \|\boldsymbol{\mu}_{v,t}^{(l)} - \mathbb{E}[\widehat{\boldsymbol{\mu}}_{v,t}^{(l)}]\|^2 + \mathbf{V}_{p_t}(\widehat{\boldsymbol{\mu}}_{v,t}^{(l)}) \triangleq \text{Bias}(\widehat{\boldsymbol{\mu}}_{v,t}^{(l)}) + \mathbf{V}_{p_t}(\widehat{\boldsymbol{\mu}}_{v,t}^{(l)}).$$

Then, by letting $k = 1$, we have:

$$\text{Bias}(\widehat{\boldsymbol{\mu}}_{v,t}^{(l)}) = D_v^2\left\| \sum_{i \in \mathcal{N}_v} p_{i,t} \boldsymbol{z}_{i,t}^{(l)} - \frac{1}{D_v} \sum_{j \in \mathcal{N}_v} \boldsymbol{z}_{j,t}^{(l)} \right\|^2$$

$$\mathbf{V}_{p_t}(\widehat{\boldsymbol{\mu}}_{v,t}^{(l)}) = D_v^2\mathbb{E}\left[\left\| \boldsymbol{z}_{i,t}^{(l)} - \sum_{j \in \mathcal{N}_v} p_{j,t} \boldsymbol{z}_{j,t}^{(l)} \right\|^2\right]$$

$$\nabla_{p_{i,t}}\text{Bias}(\widehat{\boldsymbol{\mu}}_{v,t}^{(l)}) = D_v^2\left( p_{i,t}\|\boldsymbol{z}_{i,t}^{(l)}\|^2 + \boldsymbol{z}_{i,t}^{(l)\top} \sum_{j \in \mathcal{N}_v} p_{j,t} \boldsymbol{z}_{j,t}^{(l)} - 2\boldsymbol{z}_{i,t}^{(l)\top} \sum_{j \in \mathcal{N}_v} \frac{\boldsymbol{z}_{j,t}^{(l)}}{D_v} \right)$$

$$\nabla_{p_{i,t}}\mathbf{V}_{p_t} = D_v^2\left( (1 - p_{i,t})\|\boldsymbol{z}_{i,t}^{(l)}\|^2 - \boldsymbol{z}_{i,t}^{(l)\top} \sum_{j \in \mathcal{N}_v} p_{j,t} \boldsymbol{z}_{j,t}^{(l)} \right)$$

In an effort to find an improved balance for bias-variance trade-off, we optimize the bias and variance simultaneously, defining the reward as the negative gradient w.r.t. both terms:

$$r_{i,t} = -\frac{1}{D_v^2}\nabla_{p_{i,t}}\text{Bias}(\boldsymbol{\mu}_{v,t}^{(l)}) - \frac{1}{D_v^2}\nabla_{p_{i,t}}\mathbf{V}_{p_t}(\boldsymbol{\mu}_{v,t}^{(l)})$$

$$= 2\boldsymbol{z}_{i,t}^{(l)\top}\bar{\boldsymbol{z}}_{v,t}^{(l)} - \left\|\boldsymbol{z}_{i,t}^{(l)}\right\|^2$$

(11)

where $\bar{\boldsymbol{z}}_{v,t}^{(l)} = \frac{1}{D_v}\boldsymbol{\mu}_{v,t}^{(l)} = \frac{1}{D_v}\sum_{j \in \mathcal{N}_v} \boldsymbol{z}_{j,t}^{(l)}$. Note that our reward assigns the same weight to the gradients of bias and variance terms. Actually, we can in principle scale the two gradients differently

to explore a different balance between bias and variance. For example, deriving the reward with $k > 1$ is equivalent to weighting variance lower than bias. We leave it as a future work.

# E  The Proof of Variation Budget

**Lemma** 1 (Dynamic of Embedding) *Based on our assumptions on GCN, for any $i \in \mathcal{V}$ at the layer $l$, we have:*

$$\left\| \boldsymbol{z}_{i,t}^{(l)} \right\| \leq C_z, \quad \left| \tilde{r}_{i,t} \right| \leq C_r, \quad \left| r_{i,t} \right| \leq C_r, \tag{12}$$

*where $C_z = G^{l-1}\bar{A}C_\sigma C_\theta C_x$ and $C_r = 3C_z^2$. Then, consider the training dynamic of GCN optimized by SGD. For any node $i \in \mathcal{V}$ at the layer $l$, we have*

$$\Delta_{t,l}^z = \max_{i \in \mathcal{V}} \left\| \boldsymbol{z}_{i,t+1}^{(l)} - \boldsymbol{z}_{i,t}^{(l)} \right\| \leq \alpha_t G^{l-1}\bar{A}C_\sigma C_x C_g.$$

*Proof.* We have to clarify that a typo was found in (6) after submitting the full paper. (12) is the correct version of (6) and the other proofs actually depend on (12) and still hold true after switching (6) to (12). We will correct this typo in the final version. Let $v = \text{argmax}_{i \in \mathcal{V}}\left(\|\boldsymbol{h}_{i,t}^{(l)}\|\right)$, for any $1 \leq t \leq T$

$$\|\boldsymbol{h}_{v,t}^{(l)}\| = \left\|\sigma\Big(\sum_{i \in \mathcal{N}_v} a_{vi}\boldsymbol{h}_{i,t}^{(l-1)}W_t^{(l-1)}\Big) - \sigma\Big(\boldsymbol{0}\Big)\right\| \leq C_\sigma \|\sum_{i \in \mathcal{N}_v} a_{vi}\boldsymbol{h}_{i,t}^{(l-1)}W_t^{(l-1)}\| \tag{13}$$

$$\leq C_\sigma \|\sum_{i \in \mathcal{N}_v} a_{vi}\boldsymbol{h}_{i,t}^{(l-1)}\|\|W_t^{(l-1)}\| \leq C_\sigma C_\theta \sum_{i \in \mathcal{N}_v} |a_{vi}|\|\boldsymbol{h}_{i,t}^{(l-1)}\| \tag{14}$$

$$\leq C_\sigma C_\theta \bar{A}\bar{D} \max_{i \in \mathcal{V}} \|\boldsymbol{h}_{i,t}^{(l-1)}\| \tag{15}$$

$$\leq (C_\sigma C_\theta \bar{A}\bar{D})^{l-1} \max_{i \in \mathcal{V}} \|\boldsymbol{h}_{i,t}^{(1)}\| \tag{16}$$

$$\leq (C_\sigma C_\theta \bar{A}\bar{D})^{l-1} C_\sigma C_\theta \max_{j \in \mathcal{V}} \|\sum_{i \in \mathcal{N}_j} a_{ji}\boldsymbol{x}_i\| \tag{17}$$

$$\leq (C_\sigma C_\theta \bar{A}\bar{D})^{l-1} C_\sigma C_\theta C_x \tag{18}$$

$$\leq G^{l-1} C_\sigma C_\theta C_x \tag{19}$$

(13) uses the assumption that the activation function $\sigma(\cdot)$ is $C_\sigma$-Lipschitz continuous function. (14) uses the assumption of bounded parameters and the triangle inequality. (16) recursively expands (15) from layer-$l$ to 1-st layer. Therefore, we can obtain, for any $v \in \mathcal{V}$, any $1 \leq t \leq T$

$$\|\boldsymbol{h}_{v,t}^{(l)}\| \leq G^{l-1} C_\sigma C_\theta C_x, \quad \text{and} \quad \|\boldsymbol{z}_{i,t}^{(l)}\| \leq G^{l-1}\bar{A}C_\sigma C_\theta C_x = C_z. \tag{20}$$

Then, define $\Delta_{t,l}^h = \max_{i \in \mathcal{V}} \|\boldsymbol{h}_{i,t+1}^{(l)} - \boldsymbol{h}_{i,t}^{(l)}\|$, for any $v \in \mathcal{V}$,

$$\|\boldsymbol{h}_{v,t+1}^{(l+1)} - \boldsymbol{h}_{v,t}^{(l+1)}\| = \left\|\sigma\Big(\sum_{i \in \mathcal{N}_v} a_{vi}\boldsymbol{h}_{i,t+1}^{(l)}W_{t+1}^{(l)}\Big) - \sigma\Big(\sum_{i \in \mathcal{N}_v} a_{vi}\boldsymbol{h}_{i,t}^{(l)}W_t^{(l)}\Big)\right\| \tag{21}$$

$$\leq C_\sigma \left\|\sum_{i \in \mathcal{N}_v} a_{vi}\boldsymbol{h}_{i,t+1}^{(l)}\Big(W_t^{(l)} - \alpha_t \nabla_{W_t^{(l)}}\widehat{\mathcal{L}}\Big) - \sum_{i \in \mathcal{N}_v} a_{vi}\boldsymbol{h}_{i,t}^{(l)}W_t^{(l)}\right\| \tag{22}$$

$$\leq C_\sigma \|\sum_{i \in \mathcal{N}_v} a_{vi}(\boldsymbol{h}_{i,t+1}^{(l)} - \boldsymbol{h}_{i,t}^{(l)})W_t^{(l)}\| + C_\sigma \alpha_t \|\sum_{i \in \mathcal{N}_v} a_{vi}\boldsymbol{h}_{i,t+1}^{(l)}\nabla_{W_t^{(l)}}\widehat{\mathcal{L}}\| \tag{23}$$

$$\leq C_\sigma \|\sum_{i \in \mathcal{N}_v} a_{vi}(\boldsymbol{h}_{i,t+1}^{(l)} - \boldsymbol{h}_{i,t}^{(l)})\|\|W_t^{(l)}\| + \alpha_t C_\sigma \|\sum_{i \in \mathcal{N}_v} a_{vi}\boldsymbol{h}_{i,t+1}^{(l)}\|\|\nabla_{W_t^{(l)}}\widehat{\mathcal{L}}\| \tag{24}$$

$$\leq C_\sigma C_\theta \bar{A} \sum_{i \in \mathcal{N}_v} \|\boldsymbol{h}_{i,t+1}^{(l)} - \boldsymbol{h}_{i,t}^{(l)}\| + \alpha_t C_\sigma \sum_{i \in \mathcal{N}_v} |a_{vi}|\|\boldsymbol{h}_{i,t+1}^{(l)}\|\|\nabla_{W_t^{(l)}}\widehat{\mathcal{L}}\| \tag{25}$$

$$\leq C_\sigma C_\theta \bar{A}\bar{D} \max_{i \in \mathcal{V}} \|\boldsymbol{h}_{i,t+1}^{(l)} - \boldsymbol{h}_{i,t}^{(l)}\| + \alpha_t \|\nabla_{W_t^{(l)}}\widehat{\mathcal{L}}\| \cdot C_\sigma \bar{A}\bar{D} \max_{i \in \mathcal{V}} \|\boldsymbol{h}_{i,t+1}^{(l)}\| \tag{26}$$

$$\leq G\Delta_{t,l}^h + \alpha_t \|\nabla_{W_t^{(l)}}\widehat{\mathcal{L}}\| \cdot C_\sigma \bar{A}\bar{D}C_\sigma C_\theta (C_\sigma C_\theta \bar{A}\bar{D})^{l-1} C_x \tag{27}$$

$$= G\Delta_{t,l}^h + \alpha_t G^l C_\sigma C_x \|\nabla_{W_t^{(l)}}\widehat{\mathcal{L}}\| \tag{28}$$

i.e.

$$\Delta_{t,l+1}^h \le G\Delta_{t,l}^h + \alpha_t G^l C_\sigma C_x \|\nabla_{W_t^{(l)}}\widehat{\mathcal{L}}\| \tag{29}$$

$$\Delta_{t,l}^h \le G\Delta_{t,l-1}^h + \alpha_t G^{l-1} C_\sigma C_x \|\nabla_{W_t^{(l-1)}}\widehat{\mathcal{L}}\| \tag{30}$$

(22) uses the update rule of SGD and the $C_\sigma$-Lipschitz continuous assumption of $\sigma(\cdot)$. (27) is based on (20). Then, we recursively unfold the above inequality from layer-$l$ to 1-st layer:

$$\begin{aligned}
\Delta_{t,l}^h &\le G\Delta_{t,l-1}^h + \alpha_t G^{l-1} C_\sigma C_x \|\nabla_{W_t^{(l-1)}}\widehat{\mathcal{L}}\| \\
&\le G\Big(G\Delta_{t,l-2}^h + \alpha_t G^{l-2} C_\sigma C_x \|\nabla_{W_t^{(l-2)}}\widehat{\mathcal{L}}\|\Big) + \alpha_t G^{l-1} C_\sigma C_x \|\nabla_{W_t^{(l-1)}}\widehat{\mathcal{L}}\| \\
&\le G^2 \Delta_{t,l-2}^h + \alpha_t G^{l-1} C_\sigma C_x \Big(\|\nabla_{W_t^{(l-1)}}\widehat{\mathcal{L}}\| + \|\nabla_{W_t^{(l-2)}}\widehat{\mathcal{L}}\|\Big) \\
&\quad \cdots \\
&\le G^{l-1}\Delta_{t,1}^h + \alpha_t G^{l-1} C_\sigma C_x \sum_{j=1}^{l-1}\|\nabla_{W_t^{(j)}}\widehat{\mathcal{L}}\| \\
&\le G^{l-1} C_\sigma \|\sum_{i\in\mathcal{N}_{v_1}} a_{v_1 i}\boldsymbol{x}_i\|\|W_{t+1}^{(0)} - W_t^{(0)}\| + \alpha_t G^{l-1} C_\sigma C_x \sum_{j=1}^{l-1}\|\nabla_{W_t^{(j)}}\widehat{\mathcal{L}}\| \\
&\le \alpha_t G^{l-1} C_\sigma C_x \|\nabla_{W_t^{(0)}}\widehat{\mathcal{L}}\| + \alpha_t G^{l-1} C_\sigma C_x \sum_{j=1}^{l-1}\|\nabla_{W_t^{(j)}}\widehat{\mathcal{L}}\| \\
&= \alpha_t G^{l-1} C_\sigma C_x \sum_{j=0}^{l-1}\|\nabla_{W_t^{(j)}}\widehat{\mathcal{L}}\| \le \alpha_t G^{l-1} C_\sigma C_x C_g
\end{aligned}$$

Therefore,

$$\Delta_{t,l}^h = \max_{i\in\mathcal{V}}\|\boldsymbol{h}_{i,t+1}^{(l)} - \boldsymbol{h}_{i,t}^{(l)}\| \le \alpha_t G^{l-1} C_\sigma C_x C_g \tag{31}$$

Since $a_{vi}$ is fixed in GCN, we can further get

$$\Delta_{t,l}^z = \max_{i\in\mathcal{V}}\left\|\boldsymbol{z}_{i,t+1}^{(l)} - \boldsymbol{z}_{i,t}^{(l)}\right\| \le \bar{A}\Delta_{t,l}^h \le \alpha_t G^{l-1}\bar{A}C_\sigma C_x C_g. \tag{32}$$

Meanwhile, for any $i\in\mathcal{V}$, we have

$$\begin{aligned}
|\tilde{r}_{i,t}| &= \left|\mathrm{ReLU}\Big(2\boldsymbol{z}_{i,t}^{(l)\top}\sum_{j\in\mathcal{S}_t}\frac{1}{k}\boldsymbol{z}_{j,t}^{(l)} - \|\boldsymbol{z}_{i,t}^{(l)}\|^2\Big)\right| \\
&\le \left|2\boldsymbol{z}_{i,t}^{(l)\top}\sum_{j\in\mathcal{S}_t}\frac{1}{k}\boldsymbol{z}_{j,t}^{(l)} - \|\boldsymbol{z}_{i,t}^{(l)}\|^2\right| \le 2\left|\boldsymbol{z}_{i,t}^{(l)\top}\sum_{j\in\mathcal{S}_t}\frac{1}{k}\boldsymbol{z}_{j,t}^{(l)}\right| + \left\|\boldsymbol{z}_{i,t}^{(l)}\right\|^2 \\
&\le 2\left\|\boldsymbol{z}_{i,t}^{(l)}\right\|\left\|\sum_{j\in\mathcal{S}_t}\frac{1}{k}\boldsymbol{z}_{j,t}^{(l)}\right\| + \left\|\boldsymbol{z}_{i,t}^{(l)}\right\|^2 \\
&\le 2\left\|\boldsymbol{z}_{i,t}^{(l)}\right\| \cdot \frac{1}{k}\sum_{j\in\mathcal{S}_t}\left\|\boldsymbol{z}_{j,t}^{(l)}\right\| + \left\|\boldsymbol{z}_{i,t}^{(l)}\right\|^2 \le 3C_z^2
\end{aligned}$$

The first inequality uses the fact that $\mathrm{ReLU}(\cdot)$ is 1-Lipschitz continuous and $\mathrm{ReLU}(0) = 0$. Similarly, for $r_{i,t}$ in (3), we have:

$$\begin{aligned}
|r_{i,t}| &\le 2\left|\boldsymbol{z}_{i,t}^{(l)\top}\sum_{j\in\mathcal{N}_v}\frac{1}{D_v}\boldsymbol{z}_{j,t}^{(l)}\right| + \left\|\boldsymbol{z}_{i,t}^{(l)}\right\|^2 \\
&\le 2\left\|\boldsymbol{z}_{i,t}^{(l)}\right\|\left\|\sum_{j\in\mathcal{N}_v}\frac{1}{D_v}\boldsymbol{z}_{j,t}^{(l)}\right\| + \left\|\boldsymbol{z}_{i,t}^{(l)}\right\|^2 \\
&\le 2\left\|\boldsymbol{z}_{i,t}^{(l)}\right\| \cdot \frac{1}{D_v}\sum_{j\in\mathcal{N}_v}\left\|\boldsymbol{z}_{j,t}^{(l)}\right\| + \left\|\boldsymbol{z}_{i,t}^{(l)}\right\|^2 \le 3C_z^2.
\end{aligned}$$

$\square$

**Lemma** 2 (Variation Budget) *Given the learning rate schedule of SGD as $\alpha_t = 1/t$ and our assumptions on the GCN training, for any $T \ge 2$, any $v \in \mathcal{V}$, the variation of the expected reward in*

(3) *and* (4) *can be bounded as:*

$$\sum_{t=1}^{T} \left| \mathbb{E}[r_{i,t+1}] - \mathbb{E}[r_{i,t}] \right| \leq V_T = \bar{C}_v \ln T, \quad \sum_{t=1}^{T} \left| \mathbb{E}[\tilde{r}_{i,t+1}] - \mathbb{E}[\tilde{r}_{i,t}] \right| \leq V_T = \bar{C}_v \ln T$$

*where* $\bar{C}_v = 12 G^{2(l-1)} C_\sigma^2 C_x^2 \bar{A}^2 C_\theta C_g$.

*Proof.* Based on Lemma. 1, we then discuss the variation budget of our reward $\tilde{r}_{i,t} = \text{ReLU}\left(2 z_{i,t}^{(l)\top} \bar{z}_{v,t}^{(l)} - \|z_{i,t}^{(l)}\|^2\right)$, where $\bar{z}_{v,t}^{(l)} = \frac{1}{k} \sum_{j \in \mathcal{S}_t} z_{j,t}^{(l)}$. For any $v \in \mathcal{V}$, any $i \in \mathcal{N}_v$,

$$
\begin{align}
\left| \tilde{r}_{i,t+1} - \tilde{r}_{i,t} \right| =& \left| \text{ReLU}(2 z_{i,t+1}^{(l)\top} \bar{z}_{v,t+1}^{(l)} - \|z_{i,t+1}^{(l)}\|^2) - \text{ReLU}(2 z_{i,t}^{(l)\top} \bar{z}_{v,t}^{(l)} - \|z_{i,t}^{(l)}\|^2) \right| \tag{33} \\
\leq& \left| (2 z_{i,t+1}^{(l)\top} \bar{z}_{v,t+1}^{(l)} - \|z_{i,t+1}^{(l)}\|^2) - (2 z_{i,t}^{(l)\top} \bar{z}_{v,t}^{(l)} - \|z_{i,t}^{(l)}\|^2) \right| \tag{34} \\
\leq& 2 \left| z_{i,t+1}^{(l)\top} \bar{z}_{v,t+1}^{(l)} - z_{i,t}^{(l)\top} \bar{z}_{v,t}^{(l)} \right| + \left| \|z_{i,t+1}^{(l)}\|^2 - \|z_{i,t}^{(l)}\|^2 \right| \tag{35} \\
\leq& 2 \left| z_{i,t+1}^{(l)\top} \bar{z}_{v,t+1}^{(l)} - z_{i,t}^{(l)\top} \bar{z}_{v,t+1}^{(l)} \right| + 2 \left| z_{i,t}^{(l)\top} \bar{z}_{v,t+1}^{(l)} - z_{i,t}^{(l)\top} \bar{z}_{v,t}^{(l)} \right| \tag{36} \\
& + \left| (z_{i,t+1}^{(l)} - z_{i,t}^{(l)})^\top z_{i,t+1}^{(l)} \right| + \left| z_{i,t}^{(l)\top} (z_{i,t+1}^{(l)} - z_{i,t}^{(l)}) \right| \tag{37} \\
\leq& 2 \left\| z_{i,t+1}^{(l)} - z_{i,t}^{(l)} \right\| \left\| \bar{z}_{v,t+1}^{(l)} \right\| + 2 \left\| z_{i,t}^{(l)} \right\| \left\| \bar{z}_{v,t+1}^{(l)} - \bar{z}_{v,t}^{(l)} \right\| \tag{38} \\
& + \left\| z_{i,t+1}^{(l)} - z_{i,t}^{(l)} \right\| \left\| z_{i,t+1}^{(l)} \right\| + \left\| z_{i,t}^{(l)} \right\| \left\| z_{i,t+1}^{(l)} - z_{i,t}^{(l)} \right\| \tag{39} \\
\leq& 2 \Delta_{t,l}^z \cdot \frac{1}{k} \sum_{j \in \mathcal{S}_t} \left\| z_{j,t+1}^{(l)} \right\| + 2 C_z \cdot \frac{1}{k} \sum_{j \in \mathcal{S}_t} \left\| z_{j,t+1}^{(l)} - z_{j,t}^{(l)} \right\| \tag{40} \\
& + \Delta_{t,l}^z \cdot C_z + C_z \cdot \Delta_{t,l}^z \tag{41} \\
\leq& 6 C_z \Delta_{t,l}^z \tag{42} \\
=& 6 G^{2(l-1)} C_\sigma^2 \bar{A}^2 C_x^2 C_g C_\theta \cdot \alpha_t \tag{43}
\end{align}
$$

(34) uses the fact that ReLU is 1-Lipschitz continuous function. (39) uses the Cauchy-Schwarz inequality. (41) uses the triangle inequality. Then, given the expected reward $\mathbb{E}[\tilde{r}_{i,t}] = \sum_{\mathcal{S}_1,\ldots,\mathcal{S}_{t-1}} p(\mathcal{S}_1, \ldots, \mathcal{S}_{t-1}) \tilde{r}_{i,t}$, we can obtain that for any $v \in \mathcal{V}$, any $i \in \mathcal{N}_v$ and any $1 \leq t \leq T$,

$$
\begin{align}
\left| \mathbb{E}[\tilde{r}_{i,t+1}] - \mathbb{E}[\tilde{r}_{i,t}] \right| =& \left| \sum_{\mathcal{S}_1, \mathcal{S}_2, \ldots, \mathcal{S}_t} p(\mathcal{S}_1, \ldots, \mathcal{S}_t) \tilde{r}_{i,t+1} - \sum_{\mathcal{S}_1, \mathcal{S}_2, \ldots, \mathcal{S}_{t-1}} p(\mathcal{S}_1, \ldots, \mathcal{S}_{t-1}) \tilde{r}_{i,t} \right| \tag{44} \\
=& \left| \sum_{\mathcal{S}_1, \ldots, \mathcal{S}_{t-1}} p(\mathcal{S}_1, \ldots, \mathcal{S}_{t-1}) \sum_{\mathcal{S}_t} p(\mathcal{S}_t | \mathcal{S}_1, \ldots, \mathcal{S}_{t-1}) \left( \tilde{r}_{i,t+1} - \tilde{r}_{i,t} \right) \right| \tag{45} \\
=& \sum_{\mathcal{S}_1, \ldots, \mathcal{S}_{t-1}} p(\mathcal{S}_1, \ldots, \mathcal{S}_{t-1}) \sum_{\mathcal{S}_t} p(\mathcal{S}_t | \mathcal{S}_1, \ldots, \mathcal{S}_{t-1}) \left| \tilde{r}_{i,t+1} - \tilde{r}_{i,t} \right| \tag{46} \\
\leq& \sum_{\mathcal{S}_1, \ldots, \mathcal{S}_{t-1}} p(\mathcal{S}_1, \ldots, \mathcal{S}_{t-1}) \sum_{\mathcal{S}_t} p(\mathcal{S}_t | \mathcal{S}_1, \ldots, \mathcal{S}_{t-1}) 6 C_z \Delta_{t,l}^z \tag{47} \\
=& 6 C_z \Delta_{t,l}^z = 6 G^{2(l-1)} C_\sigma^2 \bar{A}^2 C_x^2 C_g C_\theta \cdot \alpha_t \tag{48}
\end{align}
$$

(45) uses the fact that $\tilde{r}_{i,t}$ is only a function of previous actions $(\mathcal{S}_1, \mathcal{S}_2, \ldots, \mathcal{S}_{t-1})$ and does not depends on $\{\mathcal{S}_\tau | \tau \geq t\}$. Therefore, we have the variation budget of expected rewards as

$$
\begin{align*}
\sum_{t=1}^{T} \sup \left| \mathbb{E}[\tilde{r}_{i,t+1}] - \mathbb{E}[\tilde{r}_{i,t}] \right| \leq& \sum_{t=1}^{T} 6 G^{2(l-1)} C_\sigma^2 \bar{A}^2 C_x^2 C_g C_\theta \cdot \alpha_t \\
=& 6 G^{2(l-1)} C_\sigma^2 \bar{A}^2 C_x^2 C_g C_\theta \sum_{t=1}^{T} \frac{1}{t} \\
=& 6 G^{2(l-1)} C_\sigma^2 \bar{A}^2 C_x^2 C_g C_\theta (\ln T + \epsilon)
\end{align*}
$$

where $\epsilon$ is the Euler-Mascheroni constant. Hence, for $T \geq 2$, we have,

$$\sum_{t=1}^{T} \sup \left| \mathbb{E}[\tilde{r}_{i,t+1}] - \mathbb{E}[\tilde{r}_{i,t}] \right| \leq \bar{C}_v \ln T \tag{49}$$

where $\bar{C}_v = 12G^{2(l-1)}C_\sigma^2 \bar{A}^2 C_x^2 C_g C_\theta$.

Then, for the reward function in 3, we also have

$$
\begin{aligned}
\left| \tilde{r}_{i,t+1} - \tilde{r}_{i,t} \right| =& \left| (2\boldsymbol{z}_{i,t+1}^{(l)\top} \bar{\boldsymbol{z}}_{v,t+1}^{(l)} - \|\boldsymbol{z}_{i,t+1}^{(l)}\|^2) - (2\boldsymbol{z}_{i,t}^{(l)\top} \bar{\boldsymbol{z}}_{v,t}^{(l)} - \|\boldsymbol{z}_{i,t}^{(l)}\|^2) \right| \\
\leq& 2 \left| \boldsymbol{z}_{i,t+1}^{(l)\top} \bar{\boldsymbol{z}}_{v,t+1}^{(l)} - \boldsymbol{z}_{i,t}^{(l)\top} \bar{\boldsymbol{z}}_{v,t}^{(l)} \right| + \left| \|\boldsymbol{z}_{i,t+1}^{(l)}\|^2 - \|\boldsymbol{z}_{i,t}^{(l)}\|^2 \right| \\
\leq& 2 \left| \boldsymbol{z}_{i,t+1}^{(l)\top} \bar{\boldsymbol{z}}_{v,t+1}^{(l)} - \boldsymbol{z}_{i,t}^{(l)\top} \bar{\boldsymbol{z}}_{v,t+1}^{(l)} \right| + 2 \left| \boldsymbol{z}_{i,t}^{(l)\top} \bar{\boldsymbol{z}}_{v,t+1}^{(l)} - \boldsymbol{z}_{i,t}^{(l)\top} \bar{\boldsymbol{z}}_{v,t}^{(l)} \right| \\
&+ \left| (\boldsymbol{z}_{i,t+1}^{(l)} - \boldsymbol{z}_{i,t}^{(l)})^\top \boldsymbol{z}_{i,t+1}^{(l)} \right| + \left| \boldsymbol{z}_{i,t}^{(l)\top} (\boldsymbol{z}_{i,t+1}^{(l)} - \boldsymbol{z}_{i,t}^{(l)}) \right| \\
\leq& 2 \left\| \boldsymbol{z}_{i,t+1}^{(l)} - \boldsymbol{z}_{i,t}^{(l)} \right\| \left\| \bar{\boldsymbol{z}}_{v,t+1}^{(l)} \right\| + 2 \left\| \boldsymbol{z}_{i,t}^{(l)} \right\| \left\| \bar{\boldsymbol{z}}_{v,t+1}^{(l)} - \bar{\boldsymbol{z}}_{v,t}^{(l)} \right\| \\
&+ \left\| \boldsymbol{z}_{i,t+1}^{(l)} - \boldsymbol{z}_{i,t}^{(l)} \right\| \left\| \boldsymbol{z}_{i,t+1}^{(l)} \right\| + \left\| \boldsymbol{z}_{i,t}^{(l)} \right\| \left\| \boldsymbol{z}_{i,t+1}^{(l)} - \boldsymbol{z}_{i,t}^{(l)} \right\| \\
\leq& 2\Delta_{t,l}^z \cdot \frac{1}{D_v} \sum_{j \in \mathcal{N}_v} \left\| \boldsymbol{z}_{j,t+1}^{(l)} \right\| + 2C_z \cdot \frac{1}{D_v} \sum_{j \in \mathcal{N}_v} \left\| \boldsymbol{z}_{j,t+1}^{(l)} - \boldsymbol{z}_{j,t}^{(l)} \right\| + \Delta_{t,l}^z \cdot C_z + C_z \cdot \Delta_{t,l}^z \\
\leq& 6C_z \Delta_{t,l}^z = 6G^{2(l-1)}C_\sigma^2 \bar{A}^2 C_x^2 C_g C_\theta \cdot \alpha_t
\end{aligned}
$$

For its expected reward $\mathbb{E}[r_{i,t}] = \sum_{\mathcal{S}_1,\ldots,\mathcal{S}_{t-1}} p(\mathcal{S}_1,\ldots,\mathcal{S}_{t-1})r_{i,t}$, we can obtain that for any $v \in \mathcal{V}$, any $i \in \mathcal{N}_v$ and any $1 \leq t \leq T$,

$$
\begin{aligned}
\left| \mathbb{E}[r_{i,t+1}] - \mathbb{E}[r_{i,t}] \right| =& \left| \sum_{\mathcal{S}_1,\mathcal{S}_2,\ldots,\mathcal{S}_t} p(\mathcal{S}_1,\ldots,\mathcal{S}_t)r_{i,t+1} - \sum_{\mathcal{S}_1,\mathcal{S}_2,\ldots,\mathcal{S}_{t-1}} p(\mathcal{S}_1,\ldots,\mathcal{S}_{t-1})r_{i,t} \right| \\
=& \left| \sum_{\mathcal{S}_1,\ldots,\mathcal{S}_{t-1}} p(\mathcal{S}_1,\ldots,\mathcal{S}_{t-1}) \sum_{\mathcal{S}_t} p(\mathcal{S}_t|\mathcal{S}_1,\ldots,\mathcal{S}_{t-1}) \Big( r_{i,t+1} - r_{i,t} \Big) \right| \\
=& \sum_{\mathcal{S}_1,\ldots,\mathcal{S}_{t-1}} p(\mathcal{S}_1,\ldots,\mathcal{S}_{t-1}) \sum_{\mathcal{S}_t} p(\mathcal{S}_t|\mathcal{S}_1,\ldots,\mathcal{S}_{t-1}) \left| r_{i,t+1} - r_{i,t} \right| \\
\leq& \sum_{\mathcal{S}_1,\ldots,\mathcal{S}_{t-1}} p(\mathcal{S}_1,\ldots,\mathcal{S}_{t-1}) \sum_{\mathcal{S}_t} p(\mathcal{S}_t|\mathcal{S}_1,\ldots,\mathcal{S}_{t-1}) \cdot 6C_z \Delta_{t,l}^z \\
=& 6C_z \Delta_{t,l}^z = 6G^{2(l-1)}C_\sigma^2 \bar{A}^2 C_x^2 C_g C_\theta \cdot \alpha_t
\end{aligned}
$$

Further, for $T \geq 2$, we have the variation budget of expected rewards as

$$\sum_{t=1}^{T} \sup \left| \mathbb{E}[r_{i,t+1}] - \mathbb{E}[r_{i,t}] \right| \leq \sum_{t=1}^{T} 6G^{2(l-1)}C_\sigma^2 \bar{A}^2 C_x^2 C_g C_\theta \cdot \alpha_t \tag{50}$$

$$= 6G^{2(l-1)}C_\sigma^2 \bar{A}^2 C_x^2 C_g C_\theta \sum_{t=1}^{T} \frac{1}{t} \tag{51}$$

$$= 6G^{2(l-1)}C_\sigma^2 \bar{A}^2 C_x^2 C_g C_\theta (\ln T + \epsilon) \leq \bar{C}_v \ln T. \tag{52}$$

$\square$

## F    The Worst-Case Regret with a Dynamic Oracle

**Theorem 3** (Regret Bound)  *Consider* (4) *as the reward function and Algorithm 1 as the neighbor sampling algorithm for training GCN. Let* $\Delta_T = (\bar{C}_v \ln T)^{-\frac{2}{3}} (D_v \ln D_v)^{\frac{1}{3}} T^{\frac{2}{3}}$, $\eta = \sqrt{\frac{2k \ln(D_v/k)}{C_r(\exp(C_r)-1)D_v T}}$, *and* $\gamma = \min\{1, \sqrt{\frac{(\exp(C_r)-1)D_v \ln(D_v/k)}{2kC_r T}}\}$. *Given the variation budget in* (8), *for every* $T \geq D_v \geq 2$, *we have the regret bound of* (9) *as*

$$\mathcal{R}(T) \leq \bar{C}(D_v \ln D_v)^{\frac{1}{3}} \cdot (T\sqrt{\ln T})^{\frac{2}{3}}.$$

*where* $\bar{C}$ *is a absolute constant independent with* $D_v$ *and* $T$.

*Proof.* Theorem 3 is a non-trivial adaptation of Theorem 2 from [4] in the context of GCN training and multiple-play setting. We consider the following regret:

$$\mathcal{R}(T) = \sum_{t=1}^{T} \sum_{i \in \mathcal{N}_k^*} \mathbb{E}[\tilde{r}_{i,t}] - \sum_{t=1}^{T} \sum_{I_t \in \mathcal{S}_t} \mathbb{E}^{\pi}[\tilde{r}_{I_t,t}]$$

where $\mathcal{N}_k^* = \operatorname{argmax}_{\mathcal{N}_k \subset \mathcal{N}_v} \sum_{i \in \mathcal{N}_k} \mathbb{E}[\tilde{r}_{i,t}], |\mathcal{N}_k| = k$. As the current reward $\tilde{r}_{i,t}$ (4) determined by previous sampling and optimization steps, the expectation $\mathbb{E}[\tilde{r}_{i,t}]$ is taken over the randomness of rewards caused by the previous history of arm pulling (action trajectory), i.e. $\mathbb{E}[\tilde{r}_{i,t}] = \sum_{(\mathcal{S}_1, \mathcal{S}_2, \ldots, \mathcal{S}_{t-1})} p(\mathcal{S}_1, \mathcal{S}_2, \ldots, \mathcal{S}_{t-1}) \cdot \tilde{r}_{i,t}$. On the other hand, the expectation $\mathbb{E}^{\pi}[\tilde{r}_{I_t,t}]$ is taken over the joint distribution of action trajectory $(\mathcal{S}_1, \mathcal{S}_2, \ldots, \mathcal{S}_T)$ of policy $\pi$. Namely, we have

$$\begin{aligned}
\mathbb{E}^{\pi}[\tilde{r}_{I_t,t}] &= \sum_{\mathcal{S}_1, \mathcal{S}_2, \ldots, \mathcal{S}_T} p(\mathcal{S}_1, \ldots, \mathcal{S}_T) \cdot \tilde{r}_{I_t,t} \\
&= \sum_{\mathcal{S}_1, \mathcal{S}_2, \ldots, \mathcal{S}_t} p(\mathcal{S}_1, \ldots, \mathcal{S}_t) \sum_{\mathcal{S}_{t+1}, \ldots, \mathcal{S}_T} p(\mathcal{S}_{t+1}, \ldots, \mathcal{S}_T | \mathcal{S}_1, \mathcal{S}_2, \ldots, \mathcal{S}_t) \cdot \tilde{r}_{I_t,t} \\
&= \sum_{\mathcal{S}_1, \mathcal{S}_2, \ldots, \mathcal{S}_t} p(\mathcal{S}_1, \ldots, \mathcal{S}_t) \cdot \tilde{r}_{I_t,t} \sum_{\mathcal{S}_{t+1}, \ldots, \mathcal{S}_T} p(\mathcal{S}_{t+1}, \ldots, \mathcal{S}_T | \mathcal{S}_1, \mathcal{S}_2, \ldots, \mathcal{S}_t) \\
&= \sum_{\mathcal{S}_1, \mathcal{S}_2, \ldots, \mathcal{S}_t} p(\mathcal{S}_1, \ldots, \mathcal{S}_t) \cdot \tilde{r}_{I_t,t}.
\end{aligned}$$

We adopt the similar idea to prove the worst-case regret: decompose $\mathcal{R}(T)$ into the gap between two oracles and the weak regret with the static oracle.

$$\begin{aligned}
\mathcal{R}(T) = \sum_{t=1}^{T} \Bigg( \sum_{i \in \mathcal{N}_k^*} \mathbb{E}[\tilde{r}_{i,t}] - \sum_{I_t \in \mathcal{S}_t} \mathbb{E}^{\pi}[\tilde{r}_{I_t,t}] \Bigg) \\
= \Bigg( \sum_{t=1}^{T} \sum_{i \in \mathcal{N}_k^*} \mathbb{E}[\tilde{r}_{i,t}] - \max_{\mathcal{N}_k \subset \mathcal{N}_v} \sum_{t=1}^{T} \sum_{i \in \mathcal{N}_k} \mathbb{E}[\tilde{r}_{i,t}] \Bigg) + \Bigg( \max_{\mathcal{N}_k \subset \mathcal{N}_v} \sum_{t=1}^{T} \sum_{i \in \mathcal{N}_k} \mathbb{E}[\tilde{r}_{i,t}] - \sum_{t=1}^{T} \sum_{I_t \in \mathcal{S}_t} \mathbb{E}^{\pi}[\tilde{r}_{I_t,t}] \Bigg)
\end{aligned}$$

where $\mathcal{N}_k^* = \operatorname{argmax}_{\mathcal{N}_k \subset \mathcal{N}_v} \sum_{i \in \mathcal{N}_k} \mathbb{E}[\tilde{r}_{i,t}], |\mathcal{N}_k| = k$, is the dynamic oracle at each step.

First, we break the horizon $[T]$ in a sequence of batches $\mathcal{T}_1, \mathcal{T}_2, \ldots, \mathcal{T}_s$ of size $\Delta_T$ each (except possible $\mathcal{T}_s$) according to Algorithm 1. For batch $m$, we decompose its regret as:

$$\sum_{t \in \mathcal{T}_m} \Bigg( \sum_{i \in \mathcal{N}_k^*} \mathbb{E}[\tilde{r}_{i,t}] - \sum_{I_t \in \mathcal{S}_t} \mathbb{E}[\tilde{r}_{I_t,t}] \Bigg) = \underbrace{\sum_{t \in \mathcal{T}_m} \sum_{i \in \mathcal{N}_k^*} \mathbb{E}[\tilde{r}_{i,t}] - \max_{\mathcal{N}_k \subset \mathcal{N}_v} \left\{ \sum_{t \in \mathcal{T}_m} \sum_{j \in \mathcal{N}_k} \mathbb{E}[\tilde{r}_{j,t}] \right\}}_{J_{1,m}} \tag{53}$$

$$+ \underbrace{\max_{\mathcal{N}_k \subset \mathcal{N}_v} \left\{ \sum_{t \in \mathcal{T}_m} \sum_{j \in \mathcal{N}_k} \mathbb{E}[\tilde{r}_{j,t}] \right\} - \sum_{t \in \mathcal{T}_m} \sum_{I_t \in \mathcal{S}_t} \mathbb{E}^{\pi}[\tilde{r}_{I_t,t}]}_{J_{2,m}} \tag{54}$$

$J_{1,m}$ is the gap between dynamic oracle and static oracle; $J_{2,m}$ is the weak regret with the static oracle. We analyze them separately. Denote the variation of rewards along $\mathcal{T}_m$ by $V_m$, i.e. $V_m = \sum_{t \in \mathcal{T}_m} \max_{i \in [D_v]} |\mathbb{E}[\tilde{r}_{i,t+1}] - \mathbb{E}[\tilde{r}_{i,t}]|$, we note that:

$$\sum_{m=1}^{s} V_m = \sum_{m=1}^{s} \sum_{t \in \mathcal{T}_m} \max_{i \in [D_v]} \left| \mathbb{E}[\tilde{r}_{i,t+1}] - \mathbb{E}[\tilde{r}_{i,t}] \right| = V_T \tag{55}$$

Let $\mathcal{N}_{k_0}$ be the static oracle in batch $\mathcal{T}_m$, i.e. $\mathcal{N}_{k_0} = \operatorname{argmax}_{\mathcal{N}_k \subset \mathcal{N}_v} \sum_{t \in \mathcal{T}_m} \sum_{j \in \mathcal{N}_k} \mathbb{E}[\tilde{r}_{j,t}]$. Then, we have: for any $t \in \mathcal{T}_m$,

$$\sum_{i \in \mathcal{N}_k^*} \mathbb{E}[\tilde{r}_{i,t}] - \sum_{i_0 \in \mathcal{N}_{k_0}} \mathbb{E}[\tilde{r}_{i_0,t}] \le 2kV_m. \tag{56}$$

(56) holds by following arguments: otherwise, there exist a time step $t_0 \in \mathcal{T}_m$, for which $\sum_{i \in \mathcal{N}_k^*} \mathbb{E}[\tilde{r}_{i,t_0}] - \sum_{i_0 \in \mathcal{N}_{k_0}} \mathbb{E}[\tilde{r}_{i_0,t_0}] \ge 2kV_m$. If so, let $\mathcal{N}_{k_1} = \operatorname{argmax}_{\mathcal{N}_k \subset \mathcal{N}_v} \sum_{i_1 \in \mathcal{N}_k} \mathbb{E}[\tilde{r}_{i_1,t_0}]$. In

such case, for all $t \in \mathcal{T}_m$, one has:

$$\sum_{i_1 \in \mathcal{N}_{k_1}} \mathbb{E}[\tilde{r}_{i_1,t}] \geq \sum_{i_1 \in \mathcal{N}_{k_1}} (\mathbb{E}[\tilde{r}_{i_1,t_0}] - V_m) \geq \sum_{i_0 \in \mathcal{N}_{k_0}} (\mathbb{E}[\tilde{r}_{i_0,t_0}] + V_m) \geq \sum_{i_0 \in \mathcal{N}_{k_0}} \mathbb{E}[\tilde{r}_{i_0,t}] \qquad (57)$$

since $V_m$ is the maximum variation of expected rewards along batch $\mathcal{T}_m$. However, (57) contradicts the optimality of $\mathcal{N}_{k_0}$ in batch $\mathcal{T}_m$. Thus, (56) holds. Therefore, we obtain

$$J_{1,m} \leq 2kV_m\Delta_T \qquad (58)$$

As for $J_{2,m}$, according to Lemma 4, the weak regret with the static oracle incurred by Exp3.M along batch $\mathcal{T}_m$ with size $\Delta_T$, tuned by $\eta = \sqrt{\frac{2k\ln(D_v/k)}{C_r(\exp(C_r)-1)D_vT}}$ and $\gamma = \min\{1, \sqrt{\frac{(\exp(C_r)-1)D_v\ln(D_v/k)}{2kC_rT}}\}$, is bounded by $\sqrt{2kC_r(\exp(C_r)-1)}\sqrt{D_v\Delta_T\ln(D_v/k)}$. Therefore, for each $m \in \{1, 2, \ldots, s\}$, we have

$$J_{2,m} = \max_{\mathcal{N}_k \subset \mathcal{N}_v} \left\{ \sum_{t \in \mathcal{T}_m} \sum_{i \in \mathcal{N}_k} \mathbb{E}[\tilde{r}_{i,t}] \right\} - \sum_{t \in \mathcal{T}_m} \sum_{I_t \in \mathcal{S}_t} \mathbb{E}^\pi[\tilde{r}_{I_t,t}] \leq \sqrt{2kC_r(\exp(C_r)-1)}\sqrt{D_v\Delta_T\ln D_v} \tag{59}$$

(59) holds because the arm is pulled according to Exp3.M policy within batch $\mathcal{T}_m$.

Then, summing over $s = \left\lceil T/\Delta_T \right\rceil$, we have

$$\mathcal{R}(T) \leq \sum_{m=1}^{s} \left( 2kV_m\Delta_T + \sqrt{2kC_r(\exp(C_r)-1)}\sqrt{D_v\Delta_T\ln D_v} \right)$$

$$\leq 2kV_T\Delta_T + \left( \frac{T}{\Delta_T} + 1 \right) \sqrt{2kC_r(\exp(C_r)-1)}\sqrt{D_v\Delta_T\ln D_v}$$

$$\leq 2kV_T\Delta_T + \sqrt{2kC_r(e^{C_r}-1)}\sqrt{\frac{D_v\ln D_v}{\Delta_T}}T + \sqrt{2kC_r(e^{C_r}-1)}\sqrt{\Delta_T D_v\ln D_v}$$

Let $\Delta_T = (D_v\ln D_v)^{\frac{1}{3}}(T/V_T)^{\frac{2}{3}}$. We have

$$\mathcal{R}(T) \leq 2k(V_T D_v\ln D_v)^{\frac{1}{3}}T^{\frac{2}{3}} + \sqrt{2kC_r(e^{C_r}-1)}(V_T D_v\ln D_v)^{\frac{1}{3}}T^{\frac{2}{3}} \tag{60}$$

$$+ \sqrt{2kC_r(e^{C_r}-1)}(D_v\ln D_v)^{\frac{2}{3}}(T/V_T)^{\frac{1}{3}} \tag{61}$$

In (61), we consider the variation budget under the GCN training (Lemma 2): $V_T = \bar{C}_v\ln T$. Assuming $\bar{C}_v = 12C_\sigma^2 C_x^2 \bar{A}^2 C_\theta C_g \cdot G^{2(l-1)} = 12C_\sigma^2 C_x^2 \bar{A}^2 C_\theta C_g \cdot (C_\sigma C_\theta \bar{A})^{2(l-1)} \cdot \bar{D}^{2(l-1)} \geq \bar{D}$, given $2 \leq D_v \leq \bar{D}$ and $T \geq D_v \geq 2$, it is easy to conclude

$$(\bar{C}_v(\ln T)D_v\ln D_v)^{\frac{1}{3}}T^{\frac{2}{3}} \geq \frac{(D_v\ln D_v)^{\frac{2}{3}}T^{\frac{1}{3}}}{(\bar{C}_v\ln T)^{\frac{1}{3}}} \tag{62}$$

Therefore,

$$\mathcal{R}(T) \leq \left( 2\sqrt{2kC_r(e^{C_r}-1)} + 2k \right) \bar{C}_v^{\frac{1}{3}} \left( D_v\ln D_v \right)^{\frac{1}{3}} \left( T\sqrt{\ln T} \right)^{\frac{2}{3}} \tag{63}$$

where we define $\bar{C} = \left( 2\sqrt{2kC_r(e^{C_r}-1)} + 2k \right) \bar{C}_v^{\frac{1}{3}}$.

Then, we consider (3) as the reward function and bound its regret. Note that we already bounded its variation budget in (52), same as the variation budget of (4). Hence, by substituting it into (61), we can get the same regret bound of (3) as (63)

$$\mathcal{R}(T) \leq \left( 2\sqrt{2kC_r(e^{C_r}-1)} + 2k \right) \bar{C}_v^{\frac{1}{3}} \left( D_v\ln D_v \right)^{\frac{1}{3}} \left( T\sqrt{\ln T} \right)^{\frac{2}{3}} \tag{64}$$

where we define $\bar{C} = \left( 2\sqrt{2kC_r(e^{C_r}-1)} + 2k \right) \bar{C}_v^{\frac{1}{3}}$. $\qquad\qquad\square$

# G   Weak Regret with a Static Oracle

**Lemma 4** (Weak Regret). *Given the reward function $\tilde{r}_{i,t} \le C_r$, set $\eta = \sqrt{\frac{2k \ln(D_v/k)}{C_r(\exp(C_r)-1)D_v T}}$ and $\gamma = \min\{1, \sqrt{\frac{(\exp(C_r)-1)D_v \ln(D_v/k)}{2kC_r T}}\}$. Then, we have the regret bound for Exp3.M as:*

$$\widehat{\mathcal{R}}(T) = \max_{\mathcal{N}_k \subset [D_v]} \sum_{t=1}^{T} \sum_{j \in \mathcal{N}_k} \mathbb{E}[\tilde{r}_{j,t}] - \sum_{t=1}^{T} \sum_{I_t \in \mathcal{S}_t} \mathbb{E}^\pi[\tilde{r}_{I_t,t}] \le \sqrt{2kC_r(\exp(C_r)-1)}\sqrt{D_v T \ln(D_v/k)} \tag{65}$$

*where $\mathcal{N}_k$ is a subset of $[D_v]$ with $k$ elements.*

*Proof.* The techniques are similar with Theorem 2 in Uchiya et al. [31] except the scale of our reward is $\tilde{r}_{i,t} \le C_r$. Besides, we explain how to take expectation over joint distribution of $(\mathcal{S}_1, \mathcal{S}_2, \ldots, \mathcal{S}_T)$ in (78) more clearly. Let $M_t, M_t'$ denote $\sum_{i=1}^{D_v} w_{i,t}, \sum_{i=1}^{D_v} w_{i,t}'$ respectively. Then, for any $t \in [T]$,

$$\frac{M_{t+1}}{M_t} = \sum_{i \in [D_v]-U_t} \frac{w_{i,t+1}}{M_t} + \sum_{i \in U_t} \frac{w_{i,t+1}}{M_t} \tag{66}$$

$$= \sum_{i \in [D_v]-U_t} \frac{w_{i,t}}{M_t} \exp(\eta \widehat{r}_{i,t}) + \sum_{i \in U_t} \frac{w_{i,t}}{M_t} \tag{67}$$

$$\le \sum_{i \in [D_v]-U_t} \frac{w_{i,t}}{M_t} \left[ 1 + \eta \widehat{r}_{i,t} + \frac{e^{C_r}-1}{2C_r}(\eta \widehat{r}_{i,t})^2 \right] + \sum_{i \in U_t} \frac{w_{i,t}}{M_t} \tag{68}$$

$$= 1 + \frac{M_t'}{M_t} \sum_{i \in [D_v]-U_t} \frac{w_{i,t}}{M_t'} \left[ \eta \widehat{r}_{i,t} + \frac{e^{C_r}-1}{2C_r}(\eta \widehat{r}_{i,t})^2 \right] \tag{69}$$

$$= 1 + \frac{M_t'}{M_t} \sum_{i \in [D_v]-U_t} \frac{p_{i,t}/k - \gamma/D_v}{1-\gamma} \left[ \eta \widehat{r}_{i,t} + \frac{e^{C_r}-1}{2C_r}(\eta \widehat{r}_{i,t})^2 \right] \tag{70}$$

$$\le 1 + \frac{\eta}{k(1-\gamma)} \sum_{i \in [D_v]-U_t} p_{i,t} \widehat{r}_{i,t} + \frac{e^{C_r}-1}{2C_r} \frac{\eta^2}{k(1-\gamma)} \sum_{i \in [D_v]-U_t} p_{i,t} \widehat{r}_{i,t}^2 \tag{71}$$

$$\le 1 + \frac{\eta}{k(1-\gamma)} \sum_{I_t \in \mathcal{S}_t - U_t} \tilde{r}_{I_t,t} + \frac{e^{C_r}-1}{2} \frac{\eta^2}{k(1-\gamma)} \sum_{i \in [D_v]} \widehat{r}_{i,t}. \tag{72}$$

Inequality (68) uses $\exp(x) \le 1 + x + \frac{\exp(C_r)-1}{2C_r}x^2$ for $x \le C_r$. Inequality (71) holds because $\frac{M_t'}{M_t} \le 1$ and inequality (72) uses the facts that $p_{i,t}\widehat{r}_{i,t} = \tilde{r}_{i,t}$ for $i \in \mathcal{S}_t$ and $p_{i,t}\widehat{r}_{i,t} = 0$ for $i \notin \mathcal{S}_t$. Since $\ln(1+x) \le x$, we have

$$\ln \frac{M_{t+1}}{M_t} \le \frac{\eta}{k(1-\gamma)} \sum_{I_t \in \mathcal{S}_t - U_t} \tilde{r}_{I_t,t} + \frac{e^{C_r}-1}{2} \frac{\eta^2}{k(1-\gamma)} \sum_{i \in [D_v]} \widehat{r}_{i,t} \tag{73}$$

By summing over $t$, we obtain

$$\ln \frac{M_T}{M_1} \le \frac{\eta}{k(1-\gamma)} \sum_{t=1}^{T} \sum_{I_t \in \mathcal{S}_t - U_t} \tilde{r}_{I_t,t} + \frac{e^{C_r}-1}{2} \frac{\eta^2}{k(1-\gamma)} \sum_{t=1}^{T} \sum_{i \in [D_v]} \widehat{r}_{i,t} \tag{74}$$

On the other hand, we have

$$\ln \frac{M_T}{M_1} \ge \ln \frac{\sum_{j \in \mathcal{N}_k} w_{j,T}}{M_1} \ge \frac{\sum_{j \in \mathcal{N}_k} \ln w_{j,T}}{k} - \ln \frac{D_v}{k} \tag{75}$$

$$= \frac{\eta}{k} \sum_{j \in \mathcal{N}_k} \sum_{t:j \notin U_t} \widehat{r}_{j,t} - \ln \frac{D_v}{k} \tag{76}$$

(75) uses the Cauchy-Schwarz inequality:

$$\sum_{j \in \mathcal{N}_k} w_{j,T} \ge k \left( \prod_{j \in \mathcal{N}_k} w_{j,T} \right)^{1/k}.$$

(76) uses the update rule of EXP3.M: $w_{j,T} = \exp(\eta \sum_{t:j \notin U_t} \widehat{r}_{j,t})$.

Thus, from (74) and (76), we conclude:

$$\sum_{j \in \mathcal{N}_k} \sum_{t:j \notin U_t} \widehat{r}_{j,t} - \frac{k}{\eta} \ln \frac{D_v}{k} \le \frac{1}{1-\gamma} \sum_{t=1}^{T} \sum_{I_t \in \mathcal{S}_t - U_t} \tilde{r}_{I_t,t} + \frac{e^{C_r}-1}{2} \frac{\eta}{1-\gamma} \sum_{t=1}^{T} \sum_{i \in [D_v]} \widehat{r}_{i,t} \quad (77)$$

Since $\sum_{j \in \mathcal{N}_k} \sum_{t:j \in U_t} \tilde{r}_{j,t} \le \frac{1}{1-\gamma} \sum_{t=1}^{T} \sum_{i \in U_t} \tilde{r}_{i,t}$ trivially holds, we have

$$\sum_{t=1}^{T} \sum_{j \in \mathcal{N}_k} \widehat{r}_{j,t} \le \sum_{t=1}^{T} \sum_{I_t \in \mathcal{S}_t} \tilde{r}_{I_t,t} + \frac{k}{\eta} \ln \frac{D_v}{k} + \frac{e^{C_r}-1}{2} \eta \sum_{t=1}^{T} \sum_{i \in [D_v]} \widehat{r}_{i,t} + \gamma \sum_{t=1}^{T} \sum_{j \in \mathcal{N}_k} \widehat{r}_{j,t} \quad (78)$$

Then, we take expectation on both side of (78) over the joint distribution $\pi$ of action trajectory $(\mathcal{S}_1, \mathcal{S}_2, \ldots, \mathcal{S}_T)$ with $\widehat{r}_{j,t}, \tilde{r}_{I_t,t}$ as the random variable. Then,

$$\mathbb{E}^\pi[\widehat{r}_{i,t}] = \sum_{\mathcal{S}_1, \mathcal{S}_2, \ldots, \mathcal{S}_T} p(\mathcal{S}_1, \mathcal{S}_2, \ldots, \mathcal{S}_T) \cdot \widehat{r}_{i,t} \quad (79)$$

$$= \sum_{\mathcal{S}_1, \ldots, \mathcal{S}_{t-1}} p(\mathcal{S}_1, \ldots, \mathcal{S}_{t-1}) \sum_{\mathcal{S}_t} p(\mathcal{S}_t | \mathcal{S}_1, \mathcal{S}_2, \ldots, \mathcal{S}_{t-1}) \sum_{\mathcal{S}_{t+1}, \ldots, \mathcal{S}_T} p(\mathcal{S}_{t+1}, \ldots, \mathcal{S}_T | \mathcal{S}_1, \ldots, \mathcal{S}_t) \cdot \widehat{r}_{i,t} \quad (80)$$

$$= \sum_{\mathcal{S}_1, \ldots, \mathcal{S}_{t-1}} p(\mathcal{S}_1, \ldots, \mathcal{S}_{t-1}) \sum_{\mathcal{S}_t} p(\mathcal{S}_t | \mathcal{S}_1, \mathcal{S}_2, \ldots, \mathcal{S}_{t-1}) \cdot \widehat{r}_{i,t} \sum_{\mathcal{S}_{t+1}, \ldots, \mathcal{S}_T} p(\mathcal{S}_{t+1}, \ldots, \mathcal{S}_T | \mathcal{S}_1, \ldots, \mathcal{S}_t) \quad (81)$$

$$= \sum_{\mathcal{S}_1, \ldots, \mathcal{S}_{t-1}} p(\mathcal{S}_1, \ldots, \mathcal{S}_{t-1}) \sum_{\mathcal{S}_t} p(\mathcal{S}_t | \mathcal{S}_1, \mathcal{S}_2, \ldots, \mathcal{S}_{t-1}) \cdot \widehat{r}_{i,t} \quad (82)$$

$$= \sum_{\mathcal{S}_1, \ldots, \mathcal{S}_{t-1}} p(\mathcal{S}_1, \ldots, \mathcal{S}_{t-1}) \cdot \mathbb{E}_{\mathcal{S}_t}[\widehat{r}_{i,t} | \mathcal{S}_1, \ldots, \mathcal{S}_{t-1}] \quad (83)$$

(81) uses the fact that $\widehat{r}_{i,t}$ does not depends on future actions $(\mathcal{S}_{t+1}, \ldots, \mathcal{S}_T)$. Then, given DepRound (Algorithm 4) selects arm-$i$ with probability $p_{i,t}$, we have $\mathbb{E}_{\mathcal{S}_t}[\widehat{r}_{i,t} | \mathcal{S}_1, \mathcal{S}_2, \ldots, \mathcal{S}_{t-1}] = p_{i,t} \cdot \frac{\tilde{r}_{i,t}}{p_{i,t}} + (1-p_{i,t}) \cdot 0 = \tilde{r}_{i,t}$. Thus, we have

$$\mathbb{E}^\pi[\widehat{r}_{i,t}] = \sum_{\mathcal{S}_1, \ldots, \mathcal{S}_{t-1}} p(\mathcal{S}_1, \mathcal{S}_2, \ldots, \mathcal{S}_{t-1}) \cdot \tilde{r}_{i,t} = \mathbb{E}[\tilde{r}_{i,t}]. \quad (84)$$

Thus, while taking expectation on both side of (78) over the joint distribution of action trajectory, we have

$$\sum_{t=1}^{T} \sum_{j \in \mathcal{N}_k} \mathbb{E}[\tilde{r}_{j,t}] - \sum_{t=1}^{T} \sum_{I_t \in \mathcal{S}_t} \mathbb{E}^\pi[\tilde{r}_{I_t,t}] \le \frac{k}{\eta} \ln \frac{D_v}{k} + \frac{e^{C_r}-1}{2} \eta \sum_{t=1}^{T} \sum_{i \in [D_v]} \mathbb{E}[\tilde{r}_{i,t}] + \gamma \sum_{t=1}^{T} \sum_{j \in \mathcal{N}_k} \mathbb{E}[\tilde{r}_{j,t}] \quad (85)$$

$$\le \frac{k}{\eta} \ln \frac{D_v}{k} + \frac{C_r(e^{C_r}-1)}{2} \eta D_v T + \gamma C_r k T. \quad (86)$$

Let $\eta = \sqrt{\frac{2k \ln(D_v/k)}{C_r(\exp(C_r)-1) D_v T}}$ and $\gamma = \min\{1, \sqrt{\frac{(\exp(C_r)-1) D_v \ln(D_v/k)}{2k C_r T}}\}$, we have the weak regret with the static oracle as:

$$\widehat{\mathcal{R}}(T) = \max_{\mathcal{N}_k \subset [D_v]} \sum_{t=1}^{T} \sum_{j \in \mathcal{N}_k} \mathbb{E}[\tilde{r}_{j,t}] - \sum_{t=1}^{T} \sum_{I_t \in \mathcal{S}_t} \mathbb{E}^\pi[\tilde{r}_{I_t,t}] \quad (87)$$

$$\le \sqrt{2k C_r(\exp(C_r)-1)} \sqrt{D_v T \ln(D_v/k)} \quad (88)$$

$\square$

## H   Implicit Assumptions for (2)

In this section, we will explain the implicit assumptions Liu et al. [24] made to hold (2) true so we focus on the reward function defined in (1). The most crucial issue lies in taking expectation on their equation (51) in Liu et al. [24]. The only random variable while taking expectation is the action, i.e.

arm pulling. The estimated reward $\widehat{r}_{i,t}$ and the policy $p_{i,t}$ can be regard as the function of actions. If the adversary is assumed non-oblivious, the setting of GNN neighbor sampling, there should be expected reward $\mathbb{E}_{(\mathcal{S}_1,\ldots,\mathcal{S}_{t-1})}[r_{i,t}]$ instead of $r_{i,t}$ after taking expectation over joint distribution of actions for $\widehat{r}_{i,t}$ in the equation (51) of Liu et al. [24]. Since $r_{i,t}$ appears in (2), they should implicitly assume the reward distribution at time step $t$ is independent with previous neighbor sampling and optimization step, i.e. oblivious adversary. Hence, they have

$$\mathbb{E}_{(\mathcal{S}_1,\ldots,\mathcal{S}_T)}\Big[\sum_{t=1}^{T}\widehat{r}_{i,t}\Big] = \sum_{t=1}^{T}\mathbb{E}_{(\mathcal{S}_1,\ldots,\mathcal{S}_T)}[\widehat{r}_{i,t}] = \sum_{t=1}^{T}\mathbb{E}_{\mathcal{S}_t}[\widehat{r}_{i,t}] = \sum_{t=1}^{T}r_{i,t}$$

Even so, there is a second issue lying in the first term $\sum_{t=1}^{T}\sum_{j\in\mathcal{N}_v}p_{j,t}\widehat{r}_{j,t}$ of r.h.s of the equation (51) in Liu et al. [24]. Although $r_{i,t}$ might be assumed from an oblivious adversary, the policy $p_{i,t}$ as a function of previous observed rewards of sampled arms cannot be assumed independent with previous actions. Hence, taking expectation for this term, i.e. $\mathbb{E}_{(\mathcal{S}_1,\ldots,\mathcal{S}_T)}\Big[\sum_{t=1}^{T}\sum_{j\in\mathcal{N}_v}p_{j,t}\widehat{r}_{j,t}\Big]$ will be quite complicated since $p_{i,t}$ is a function of $(\mathcal{S}_1,\ldots,\mathcal{S}_{t-1})$ and $\widehat{r}_{i,t}$ is a function of $\mathcal{S}_t$ given oblivious adversary, incurring a expected policy $\mathbb{E}_{(\mathcal{S}_1,\ldots,\mathcal{S}_{t-1})}[p_{i,t}]$ instead of $p_{i,t}$. Providing non-oblivious adversary, the expectation of this term will be more complicated since $p_{i,t}$ and $r_{i,t}$ depend with each other and get intertwined while taking expectation.

Auer et al. [1], Uchiya et al. [31] avoid this issue by rewriting $\sum_{j\in\mathcal{N}_v}p_{j,t}\widehat{r}_{j,t}$ as $r_{I_t,t}$ so that $p_{j,t}$ does not emerge before taking expectation. Liu et al. [24] does not adopt this technique and encounters these issues. Furthermore, Liu et al. [24] assumed the embedding is bounded: $\|\boldsymbol{h}_{i,t}\| \leq 1, \forall i \in \mathcal{V}$ in their proof, but did not verify the sensitivity of this assumption. If $\boldsymbol{h}_{i,t}$ grows beyond 1, they implicitly assumed the variation of embedding has to be bounded in that scenario.

# I  Experimental Details

All datasets we use are public standard benchmark datasets: ogbn-arxiv, ogbn-products [18], Cora-Full [5], Chameleon [11] and Squirrel [27]. For Chameleon and Squirrel, the dataset split for train/validate/test is 0.6/0.2/0.2. For ogb datasets, the dataset split follows the default option of OGB [2] (See Table 3). For CoraFull, we select 20 nodes each class for validation set, 30 nodes each class for test set and the others for training set.

Table 3: Summary of the statistics and data split of datasets.

| Dataset | # Node | # Edges | # Classes | # Features | # Train | # Val. | # Test |
|---------|--------|---------|-----------|------------|---------|--------|--------|
| Chameleon | 2,277 | 31,371 | 5 | 2325 | 1,367 | 455 | 455 |
| Squirrel | 5,201 | 198,353 | 5 | 2,089 | 3,121 | 1,040 | 1040 |
| CoraFull | 19,793 | 130,622 | 70 | 8,710 | 16,293 | 1,400 | 2,100 |
| ogbn-arxiv | 169,343 | 1,166,243 | 40 | 128 | 90,941 | 29,799 | 48,603 |
| ogbn-products | 2,449,029 | 61,859,140 | 47 | 100 | 196,615 | 39,323 | 2,213,091 |

Table 4: The detailed sampling hyperparameters for Chameleon.

| | GCN | | | GAT | | |
|---|---|---|---|---|---|---|
| Algorithm | $\gamma$ | $\eta$ | $\Delta_T$ | $\gamma$ | $\eta$ | $\Delta_T$ |
| Thanos | 0.4 | 0.01 | 1000 | 0.4 | 0.01 | 1000 |
| BanditSampler | 0.4 | 0.01 | N/A | 0.4 | 0.01 | N/A |

Table 5: The detailed sampling hyperparameters for Squirrel.

| | GCN | | | GAT | | |
|---|---|---|---|---|---|---|
| Algorithm | $\gamma$ | $\eta$ | $\Delta_T$ | $\gamma$ | $\eta$ | $\Delta_T$ |
| Thanos | 0.2 | 0.01 | 500 | 0.4 | 0.1 | 500 |
| BanditSampler | 0.4 | 0.01 | N/A | 0.4 | 0.01 | N/A |

---

[2]https://ogb.stanford.edu/

Table 6: The detailed sampling hyperparameters for CoraFull.

|  | | GCN | | | GAT | |
| --- | --- | --- | --- | --- | --- | --- |
| Algorithm | $\gamma$ | $\eta$ | $\Delta_T$ | $\gamma$ | $\eta$ | $\Delta_T$ |
| Thanos | 0.2 | 0.01 | 2000 | 0.4 | 1 | 2000 |
| BanditSampler | 0.4 | 0.01 | N/A | 0.4 | 0.01 | N/A |

Table 7: The detailed sampling hyperparameters for ogbn-arxiv.

|  | | GCN | | | GAT | |
| --- | --- | --- | --- | --- | --- | --- |
| Algorithm | $\gamma$ | $\eta$ | $\Delta_T$ | $\gamma$ | $\eta$ | $\Delta_T$ |
| Thanos | 0.2 | 1 | 8000 | 0.2 | 0.1 | 8000 |
| BanditSampler | 0.4 | 0.01 | N/A | 0.4 | 0.01 | N/A |

Table 8: The detailed sampling hyperparameters for ogbn-products.

|  | | GCN | | | GAT | |
| --- | --- | --- | --- | --- | --- | --- |
| Algorithm | $\gamma$ | $\eta$ | $\Delta_T$ | $\gamma$ | $\eta$ | $\Delta_T$ |
| Thanos | 0.2 | 0.1 | 10000 | 0.4 | 0.1 | 10000 |
| BanditSampler | 0.4 | 0.01 | N/A | 0.4 | 0.01 | N/A |

Table 9: The detailed sampling hyperparameters for cSBM synthetic data.

|  | | GCN | |
| --- | --- | --- | --- |
| Algorithm | $\gamma$ | $\eta$ | $\Delta_T$ |
| Thanos | 0.4 | 1 | 1000 |
| BanditSampler | 0.4 | 0.01 | N/A |

Table 10: The configuration of ClusterGCN.

| Dataset | Chameleon | Squirrel | Ogbn-arxiv | CoraFull | Ogbn-products |
| --- | --- | --- | --- | --- | --- |
| Partition size | 10 | 20 | 500 | 80 | 5000 |

## J  Reward Visualization

We first show the visualization of rewards to demonstrate their numerical stability in Fig. 5.

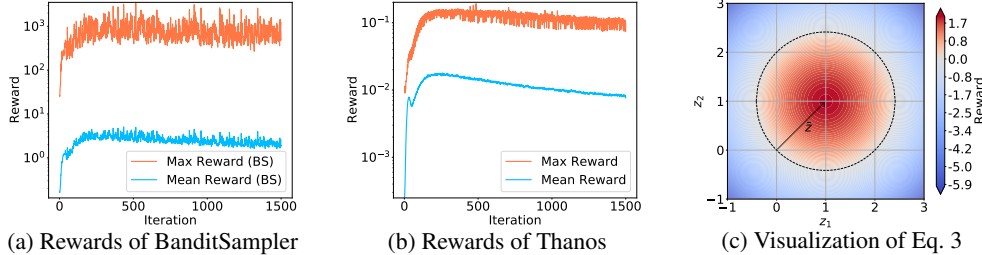

(a) Rewards of BanditSampler  (b) Rewards of Thanos  (c) Visualization of Eq. 3

Figure 5: Fig. 5a and Fig. 5b show the max vs. mean of all received rewards of two samplers during the training of GCN on Cora. Fig. 2c visualizes the reward function in (3) after setting $\bar{z}_{v,t}^{(l)} = (1,1)^\top$. Inside the dashed circle, the reward is positive, otherwise negative. When $z_{i,t}^{(l)} = \bar{z}_{v,t}^{(l)}$, it has the maximum reward.

## K  Efficiency Evaluation

To showcase the efficiency provided by our sampler, we select the ogbn-products dataset, which is sufficiently large such that loading onto a GPU is not even possible and vanilla base models like GCN

and GAT struggle. Hence we compare the time and memory usage of all methods on CPU servers. Results are shown in Table 11, where Thanos display huge gains in efficiency.

Table 11: Comparison of time and space efficiency. '#Node' denotes the number of node features involved in computation per iteration.

| | | Methods | #Node | Ave. RSS | Time/Epoch |
|---|---|---|---|---|---|
| Ogbn-products | GCN | Vanilla GCN | $1,000,700$ | 49.1GB | 24h38min |
| | | GraphSage | 2440 | 47.7GB | 499s |
| | | BanditSampler | 2462 | 47.4GB | 545s |
| | | Thanos | 2439 | **46.8GB** | **490s** |
| | GAT | Vanilla GAT | 1,010,200 | 52.5GB | 31h50min |
| | | GraphSage | 2417 | 49.8GB | **568s** |
| | | BanditSampler | 2415 | 48.7GB | 619s |
| | | Thanos | 2421 | **48.2GB** | 584s |

## L    Experiment Extension

We present the extensive experiments on cSBM in this section. Fig. 6a plots $p_{\text{intra}}$ over $\mathcal{V}_2 \cap \mathcal{V}_{\text{train}}$, which suggests both samplers have close $p_{\text{intra}}$ on $\mathcal{V}_2$.

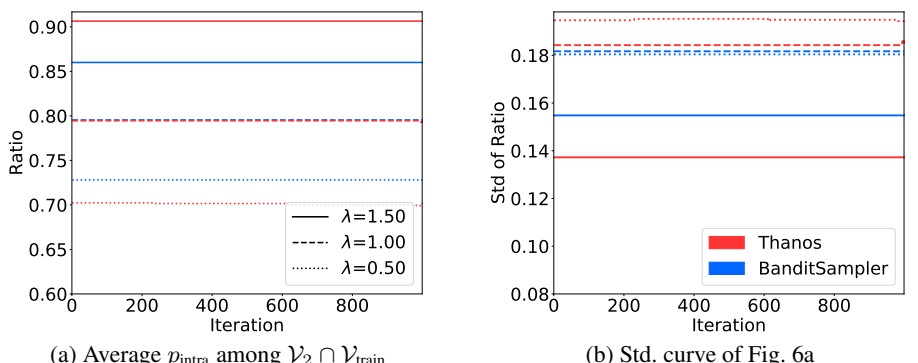

(a) Average $p_{\text{intra}}$ among $\mathcal{V}_2 \cap \mathcal{V}_{\text{train}}$    (b) Std. curve of Fig. 6a

Figure 6: Fig. 6a plots $p_{\text{intra}}$ on $\mathcal{V}_2 \cap \mathcal{V}_{\text{train}}$ and Fig. 6b plots its std. curve over $\mathcal{V}_2 \cap \mathcal{V}_{\text{train}}$.

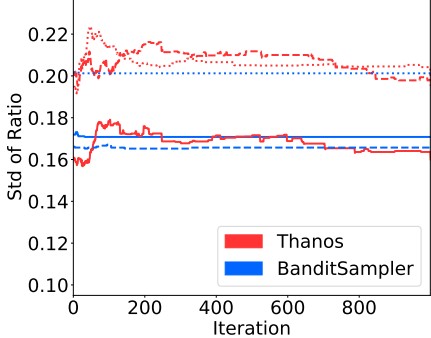

Figure 7: The Std. curve of Fig. 2a over $\mathcal{V}_1 \cap \mathcal{V}_{\text{train}}$.