# OpenReview forum: "A Biased Graph Neural Network Sampler with Near-Optimal Regret"
_NeurIPS.cc/2021/Conference — NeurIPS 2021 Poster_

### Official Review · Reviewer_eLzM · 2021-07-15

**Rating:** 7
**Confidence:** 3

**Summary:**

This paper proposes an adaptive neighbor sampling method for graph neural networks applied to large graphs. This method is based on Rexp3, an adversarial bandit algorithm. Central to it is a novel reward, which unlike previous work induces a biased sampler, which is shown to have a near optimal regret. Even though it is biased, this sampler makes up for it by having better numerical stability, requiring less unrealistic assumption for the bound, and having provably faster convergence to variance reduction of the estimator.

Empirically, the paper confirms that the behavior of the proposed method matches theoretical predictions and performs better than a more naive bandit approach on a toy graph problem and Cora. Large scale experiments are then conducted on several standard benchmarks where the proposed methods appears to outperform baselines.

**Limitations And Societal Impact:**

There are no mentions of potential negative societal impacts, but this is fairly abstract work. As an exercise, the authors may want to situate the impact of large scale graph neural networks on society.

**Main Review:**

This paper tackles a relatively specific problem in GNNs: it upgrades an existing Bandit component by simply proposing a new reward scheme. This reward scheme is very well motivated -- this is perhaps the strongest part of the paper -- and as far as I know is entirely novel.

Empirically, it is also very valuable that the predictions made by looking at the theory are verified. One problem there is that it seems all these experiments are run with only one seed; it is important to use multiple runs to be able to make stronger quantitative statements.

In terms of benchmark datasets, while it seems correct that this method beats its baselines, the magnitude of improvement seems fairly limited. I am also a bit surprised by the small magnitude of the standard deviation (assuming this is what is reported in Table 2), and since the number of seeds is not specified, it's not clear to me that the improvement is statistically significant in some cases.

The paper was fairly well written and easy to understand, although it felt very dense and with some proof-reading it seems possible to improve clarity.

In terms of significance, this paper offers a well motivated improvement to GNNs, which improves performance on benchmarks somewhat. It shows that the desire for unbiasedness in samplers may be unnecessary, and that previously proposed methods are not robust to certain issues. While I think this paper is unlikely to have a large impact, it may help in practice and offers an interesting direction for future work on this problem.

Comments:
- This might be a flaw in all samplers, I'm not familiar enough with the literature to know, but is it possible that considering only the variance of the embedding estimator causes learning issues? The backward pass is not considered, which possibly induces undesireable bias in the model, as some neighbors may have a small forward pass contribution but require a large credit assignment.
- l.74 "Recursive neighborhood expansion will cover a large portion of the graph if the graph is dense or scale-free even within a few layers." I'm not sure I understand what is being claimed here. Is this a computational argument or a variance argument? I understand that error will compound after several layers, but computationally there is no two-hop query of neighbours/no graph recursion.
- l.117 "especially in the case that the neighbors" -> "especially when neighbors"
- l.120 "to choose quite small temperature" -> "to choose a quite small temperature"
- l.138 "independent with" -> "independent from"
- l.145 This was a bit hard to parse: "we are the first to explicitly account for training dynamic in deriving reward variation and further regret bound in non-oblivious setting, and without this consideration no meaningful bound can possibly exist" I'd suggest writing e.g. "we are the first to explicitly account for training dynamics when deriving the reward variation and the improved regret bound. Without this consideration that we are in the non-oblivious setting, no meaningful bound can possibly exist".

**Time Spent Reviewing:**

4

---

> ### Author Response · Authors · 2021-08-10
> **The responses to the reviewer's questions and comments.**
>
> Thanks for the constructive feedback.
>
> **Q: Empirically, it is also very valuable that the predictions made by looking at the theory are verified. One problem there is that it seems all these experiments are run with only one seed; it is important to use multiple runs to be able to make stronger quantitative statements.**
>
> Just to clarify, we have run multiple trials and included stdev values for our main results. For example, the mean-std curves in Fig. 2(b), 2(c), comparing the sampling approximation error on Cora, are run with 10 trials. Similarly, our Table 2 results have std values in parentheses.  Beyond this, Fig 2(a) plots the approximation error on synthetic datasets with three settings of edge distributions, which reveals a manifest improvement of Thanos over BanditSampler. However, adding the std shadow in Fig.2(a) unfortunately makes this figure messy and less interpretable since there are many curves in a small space. Consequently, to make these six curves clearly visible, we chose not to include them. That being said, we agree that adding std values is a good suggestion for increasing confidence, and hence we can include them in a larger version of Fig. 2(a) placed in the revised supplementary material.
>
> **Q: In terms of benchmark datasets, while it seems correct that this method beats its baselines, the magnitude of improvement seems fairly limited. I am also a bit surprised by the small magnitude of the standard deviation (assuming this is what is reported in Table 2), and since the number of seeds is not specified, it's not clear to me that the improvement is statistically significant in some cases.**
>
> In terms of benchmark dataset, we report the mean and std with 3 trials for the large datasets (Ogbn-products, Ogbn-arxiv, CoraFull) and 5 trials for small datasets (Chameleon, Squirrel). And we found that the std is indeed not so large, indicating relatively stable performance.
>
> **Q: This might be a flaw in all samplers, I'm not familiar enough with the literature to know, but is it possible that considering only the variance of the embedding estimator causes learning issues? The backward pass is not considered, which possibly induces undesirable bias in the model, as some neighbors may have a small forward pass contribution but require a large credit assignment.**
>
> The reviewer’s intuition is a good one.  Currently, the existing samplers we are aware of only focus on forward propagation and embedding estimation. And as the reviewer suggests, this could potentially incur some degree of undesirable gradient bias during the backward pass. However, it is intractable to formulate this bias and rigorously analyze the backward pass, hence the analysis in the literature generally focuses on the forward pass as we have done.
>
> **Q: l.74 "Recursive neighborhood expansion will cover a large portion of the graph if the graph is dense or scale-free even within a few layers." I'm not sure I understand what is being claimed here. Is this a computational argument or a variance argument?**
>
> Yes, this is a computational argument. And we appreciate the reviewer’s other rephasing suggestions which can make our paper more clear. We will revise our paper according to them. Thanks.

---

> > ### Comment · Reviewer_eLzM · 2021-08-13
> > **Suggestion**
> >
> > Thank you for your responses. After reading other reviews and in light of these responses, I'd encourage the authors to find additional settings/types of graphs where their method is particularly impactful (and, for extra honesty points, where it is particularly bad). These could be (hard, doesn't have to be toy) artificially generated graphs rather than datasets. I think this may help convince readers of this paper that the method is valuable.

---

> > > ### Author Response · Authors · 2021-08-16
> > > **Response to new reviewer comment**
> > >
> > > Thanks for the additional feedback.  In this regard, the reviewer mentioned the possible inclusion of additional tests with synthetic graphs where our method is expected to be particularly impactful.  Actually, such tests can be found in our original submission.  For example, in Section 5.1 we include results using a synthetic block model, namely, the cSBM model from reference [14].  In these experiments two node clusters  $\mathcal{V}_1$ and $\mathcal{V}_2$ are generated with node features sampled from Gaussians $N_1(\mu, \sigma^2)$ and $N_2(-\mu, \sigma^2)$ respectively.  Furthermore, different edge distributions are controlled by a parameter $\lambda$.  We also scale down the node features of $\mathcal{V}_1$ by 0.1 to differentiate the distribution of feature norms and test the samplers’ sensitivity to it. From Fig. 1(b) and 2(a), we observe that Thanos outperforms the BanditSamper under different intra-class and inter-class edge distributions.  This is because Thanos is more robust to the embedding norm as predicted by our analysis.  Conversely, if there is no scaling of the features, the gap between two samplers is likely to become more similar.  Meanwhile, the experiments in Section 5.3, Figures 3(a) and 3(b), support the same conclusion on a real dataset, i.e., Thanos sampling is biased to mitigate the issue of the sensitivity to the embedding norm.
> > >
> > > And finally, just to clarify further, we note that our work is primarily of a theoretical nature aimed at providing new guarantees related to sampling behavior.  In general, this contributes to a better understanding of graph neural network sampling that is complementary to empirical studies.  Such guarantees also allow us to have more confidence in stable, reliable performance across new or unanticipated operating regimes, since any suite of benchmarks will always be of limited scope.  Even so, Thanos does consistently perform as well or better than existing methods across the multiple benchmarks we have shown, including w.r.t. new comparisons with ClusterGCN; please see our rebuttal response to Reviewer sfqg.

---

### Official Review · Reviewer_sfqg · 2021-07-16

**Rating:** 7
**Confidence:** 3

**Summary:**

This paper tackles the high variance issues of the existing sampling methods, which approximate the full-graph training of GNNs. The proposed method is built upon existing work that treats GNN neighbor sampling as a multi-armed bandit problem. The authors design a novel biased reward function to reduce variance and avoid numerical instability. The proposed reward function leads to a more meaningful notion of regret directly connected to sampling approximation error. Based on that, the authors prove that the regret of the proposed algorithm is near-optimal and demonstrate the fast convergence of the sampling approximation error. Empirical results verify that the proposed algorithm, Thanos, has improved variance estimates and performance across benchmarks over BanditSampler and others.

**Limitations And Societal Impact:**

The potential negative social impact of the work is not presented in the paper. Although the reviewer does not foresee any potential negative impacts, the authors are encouraged to add such discussions if space permits.

**Main Review:**

----------Strengths----------

(1) This work is built upon the bandit formulation of GNN sampling proposed by Liu et al. [1]. With a detailed analysis of the limitations of BanditSampler in [1] in Section 2.3, the authors propose a newly-designed reward function to address the issues. The proposed algorithm differs from the BanditSampler as it has near-optimal regret while accounting for the GNN training dynamics.

(2) The proposed algorithm, Thanos, is theoretically motivated and justified. Thanos is near-optimal, which is theoretically supported by the regret bound (Theorem 3 in the paper). Moreover, we can verify its improvement over BanditSampler in terms of sampling approximation error and final performance.

(3) Overall, the paper is well written and reasonably polished. The motivations, formulations, and theoretical analysis are presented in detail within the paper. As a small suggestion, it might be a good idea to spend a little more space on the fundamental idea of the multi-armed bandit formulation of neighbor sampling in the introduction for readers who have not heard of it before.

(4) I think the proposed algorithm and theoretical analysis are novel and valuable contributions to the multi-armed bandit problem of GNN sampling. The thoughtful analysis of the limitations of BanditSampler and the proposed solution advanced the state of the art of GNN sampling problem.

----------Weaknesses----------

(1) The authors have justified the improvement of Thanos over BanditSampler empirically. However, the empirical comparisons with other well-known sampling methods, including ClusterGCN [2] and GraphSAINT [3], are a bit insufficient, especially on models other than GCN.

(2) Moreover, I recommend the authors provide some analysis of the efficiency of the proposed algorithm in terms of both theoretical time/space complexities and empirical evaluations.

----------Overall----------

Overall, this work is an interesting and important improvement over the BanditSampler in [1]. The theoretical drawbacks of BanditSampler are fixed, and the authors prove the worst-case regret is near-optimal under mild conditions. Hence, I recommend acceptance.

----------References----------

[1] Liu, Z., Wu, Z., Zhang, Z., Zhou, J., Yang, S., Song, L. and Qi, Y., 2020. Bandit Samplers for Training Graph Neural Networks. Advances in Neural Information Processing Systems, 33.

[2] Chiang, W.L., Liu, X., Si, S., Li, Y., Bengio, S. and Hsieh, C.J., 2019, July. Cluster-gcn: An efficient algorithm for training deep and large graph convolutional networks. In Proceedings of the 25th ACM SIGKDD International Conference on Knowledge Discovery & Data Mining (pp. 257-266).

[3] Zeng, H., Zhou, H., Srivastava, A., Kannan, R. and Prasanna, V., 2019. Graphsaint: Graph sampling based inductive learning method. arXiv preprint arXiv:1907.04931.

**Time Spent Reviewing:**

5

---

> ### Author Response · Authors · 2021-08-10
> **Additional experiments suggested by the reviewer.**
>
> We appreciate your constructive feedback and add the suggested experiments.
>
> **(1) As a small suggestion, it might be a good idea to spend a little more space on the fundamental idea of the multi-armed bandit formulation of neighbor sampling in the introduction for readers who have not heard of it before.**
>
> We appreciate your helpful suggestion to add figures earlier on to illustrate MAB and neighbor sampling, which will make our paper accessible to a wider audience. We will do so in the revised version.
>
> **(2) The authors have justified the improvement of Thanos over BanditSampler empirically. However, the empirical comparisons with other well-known sampling methods, including ClusterGCN and GraphSAINT, are a bit insufficient, especially on models other than GCN.**
>
> Thanks for the constructive feedback.  We have already compared with GraphSaint in Table 2, and the original BanditSampler paper also included numerous comparisons with GraphSaint (see Table 3 in [24], reference [24] from our paper). However, ClusterGCN is indeed not included in our current submission. Even though our main contribution is on the theoretical side, adding ClusterGCN is definitely a good suggestion for broader evaluation. Hence we include here additional benchmark results on ClusterGCN under the same setup as in Table 2 from the main paper. The results are as follows:
>
> | Dataset | Chameleon | Squirrel | Ogbn-arxiv | CoraFull | Ogbn-products|
> |--|:--:|--:|--:|--:|--:|
> |Test Acc.(Std.) | 0.577(0.022) |  0.391(0.015) | 0.575(0.004) | 0.390(0.005) | 0.746(0.014)    |
> | Partition size  | 10 | 20 | 500 | 80 | 5000 |
>
> where we set the number of partitions for ClusterGCN by doing a grid search.  We can add these results to a revision.
>
> **(3) I recommend the authors provide some analysis of the efficiency of the proposed algorithm in terms of both theoretical time/space complexities and empirical evaluations.**
>
> As for the time/space complexity, our sampler is quite similar to BanditSampler. Its sampling time complexity is $O( Bk^2)$ and space complexity is $O( | \mathcal{E} | )$, where $ B $ is batch size, $k$ is sample size, and $\mathcal{E}$ is edge set. We also evaluate the complexity empirically on Ogbn-products with the same setting as Section 5.4. The following table shows the results.
>
> | Model | Methods         | #Node | Ave. RSS | Time/Epoch |
> |--|:--:|--:|--:|--:|
> | GCN | Vanilla GCN | 1,000,700 | 49.1GB   | 24h38min |
> | GCN | GraphSage      | 2440     | 47.7GB   | 499s |
> | GCN | BanditSampler | 2462     | 47.4GB   | 545s |
> | GCN | Thanos             | 2439     | 46.8GB   | 490s |
> | GAT  | Vanilla GAT | 1,010,200  | 52.5GB   | 31h50min |
> | GAT  | BanditSampler | 2415     | 48.7GB   | 610s |
> | GAT  | Thanos             | 2421     | 48.2GB   | 584s |
>
> ‘#Node’ denotes the number of node features involved in computation per iteration. Ogbn-products dataset is sufficiently large such that loading onto a GPU is not generally possible and vanilla base models like GCN and GAT struggle without sampling. Hence we compare the time and memory usage of all methods on CPU servers. Ave. RSS is the average memory usage per iteration.

---

> > ### Comment · Reviewer_sfqg · 2021-09-02
> > **Thanks for the detailed rebuttal**
> >
> > Thanks to the authors for the detailed rebuttal. I think all of my questions were addressed in a satisfactory way, and I am happy to maintain my score of 7.

---

### Official Review · Reviewer_rPqa · 2021-07-17

**Rating:** 6
**Confidence:** 2

**Summary:**

The authors propose Thanos, a bandit-based approach for subsampling graph neighbourhoods, to be used as context for graph neural network predictions. It improves on the prior art (BanditSampler) by proposing trading off increased reward bias for less variance. This allows the authors to reduce numerical optimisation issues, achieve competitive benchmarking performance, as well as explicitly take into account the GNN training dynamics in their theoretical analysis, demonstrating near-optimal regret (with an added ln T factor).

**Limitations And Societal Impact:**

No concerns.

**Main Review:**

While I found the paper very information dense and hard to follow (the lack of illustrative figures up to page 7 proved quite telling here), I was left convinced by the authors' proposed method, both in terms of claimed theoretical benefits and empirical outperformance. The baseline comparisons are relevant and demonstrate meaningful returns on strong benchmarks. All in all, the method has potential for becoming an important direction for graph subsamplers, making it relevant for a substantial amount of GNN researchers and industrial appliers of the technology.

I must stress that I am not an expert on bandits, and hence am taking most of the theoretical results at face value. My review score is primarily based on the model's empirical performance and choice of comparisons performed, and the final assessment should ideally be combined with feedback from a domain expert.

From my view, the authors build up on a previous NeurIPS paper, extend it significantly from the theory side, and then show a few expected results on standard benchmarks, which illustrate its utility for graph representation learning researchers. I think that such improvements are very useful, and therefore I suggest (weak) acceptance, and would be happy to see this paper at the conference. But the paper's key contributions are not on graph representation learning (my area of expertise) and therefore I cannot give a high confidence score. In either case I would strongly recommend the authors to revise the clarity of their method (ideally by adding more figures earlier on), and making it more accessible to non-domain-experts.

Minor typo: 'Dyanmic policy' in Table 1.

**Time Spent Reviewing:**

5

---

> ### Author Response · Authors · 2021-08-10
> **Thanks for your helpful suggestions.**
>
> We appreciate the helpful feedback and for recognizing the theoretical benefits and empirical value of our submission.  In terms of recommendations, the reviewer suggested clarifying our method (ideally by adding more figures earlier on), and making the content more accessible to non-domain-experts.  This is a good suggestion, given that ideally we would like our methodology to reach ML practitioners beyond those studying bandits and GNNs.  To the extent possible then (given space constraints, etc.), we will add figures and/or illustrations to introduce multi-armed bandits and further clarify the context of our approach.

---

### Official Review · Reviewer_R9KQ · 2021-07-19

**Rating:** 6
**Confidence:** 4

**Summary:**

The authors propose a novel bandit sampler, addressing the limitations of the previous work, for graph neural networks. The previous work utilized EXP3 which as known issues with low probabilities (variance can be really high) and leads to small learning rate. The regret bound was given in terms of a static oracle instead of a dynamic one. Also, the plays are considered in the oblivious adversary setting.
These issues are addressed in the current work by rethinking the reward function. Experimental results are shown on a wide variety of datasets as well as GNN architectures to validate the proposed algorithm.



**Limitations And Societal Impact:**

Yes.

**Main Review:**

The paper is clearly written to highlight the issues with the previous approach and the novel contributions of this work and validation via experiments.

Pros:

(A) Fixes the issues with the BanditSampler algorithm especially with regard to dynamic regret and numerical instability issues.
(B) Improvements on some of the datasets such as Chameleon and Squirrel datasets shows significant improvements in performance.

Cons/Questions:

(1) The main contribution seems to be rethinking the reward function and the rest follows by incorporating this into the BanditSampler algorithm previously introduced.
(2) How about replacing the EXP3 with an implicit version such as EXP3-IX for exploration? Will that fix the numerical issues?
(3) In some of the datasets, the improvement in performance is negligible. Can we hypothesize on which graphs would the proposed approach work better than others? Also, which is the unsampled version in the Table 2? If it is Vanilla GCN, it seems that the full averaging is worse than the sampled version but  Figure 3 shows an improvement over the sample size 'k'. So, it there an optimal selection of 'k'?

**Time Spent Reviewing:**

5

---

> ### Author Response · Authors · 2021-08-10
> **The responses to the reviewer's questions**
>
> We appreciate the constructive feedback.   Below are our responses.
>
> **Q: The main contribution seems to be rethinking the reward function and the rest follows by incorporating this into the BanditSampler algorithm previously introduced.**
>
> Besides rethinking the reward, our main contribution is in explicitly accounting for the practical training dynamics of GCN induced by SGD so as to provide a solid meaningful theoretical guarantee. This theoretical achievement addressed the limitations of prior work summarized in Section 2.3.
> Meanwhile, compared to the BanditSampler, we further consider two additional trade-offs in our algorithm: the bias-variance trade-off and the remembering-forgetting trade-off. For the latter, we dismiss the old observation by reinitializing the policy every $\Delta_T$ steps.
>
>
> **Q: How about replacing the EXP3 with an implicit version such as EXP3-IX for exploration? Will that fix the numerical issues?**
>
> EXP3-IX could perhaps partially relieve a numerically unstable policy by introducing the IX reward estimation. However, the numerical issue of BanditSampler is caused not only by reward estimation, but also because the reward itself ( $||z_i||/p_i^2$ ) is unstable due to the squared $p_i$ in the denominator. For EXP-IX, the estimated reward is formulated by $ \frac{||z_i||}{ ( p_i+\gamma ) p_i^2}$ ($\gamma$ is a constant hyperparameter). There is still $p_i^2$ in the denominator which potentially leads to an unstable policy.
>
>
> **Q: Can we hypothesize on which graphs would the proposed approach work better than others?**
>
> Our algorithm may potentially work even better, or at least be more robust, on heterophily graphs like Chameleon and Squirrel due to the intuition introduced in Section 3.1 (line 175). In this regard, the proposed reward can bias the policy towards the neighbors closer to $||\bar{z}_{v}||$, i.e. to sample the neighbors clustered together, which may help to improve the performance on heterophily graphs.
>
>
> **Q: Which is the unsampled version in Table 2? If it is Vanilla GCN, it seems that the full averaging is worse than the sampled version but Figure 3 shows an improvement over the sample size 'k'. So is there an optimal selection of 'k'?**
>
> The unsampled version is Vanilla GCN/GAT in Table 2. The neighbor sampling likely performs better than the corresponding unsampled version due to implicit regularization, analogous to DropEdge, which can sometimes help to alleviate underlying oversmoothing and overfitting issues [1]. It also depends on the dataset. For example, in our experimental results, vanilla GAT performs best on ogbn-arxiv. And the question of the optimal $k$ should generally be considered with respect to a specific computational budget. For example, increasing $k$ will increase the time/space consumption exponentially. There is also a trade-off between time/space complexity and performance, where the latter may be dependent on whether overfitting is an issue as mentioned above.
>
> [1]. Yu Rong, Wenbing Huang, Tingyang Xu, Junzhou Huang. DropEdge: Towards Deep Graph Convolutional Networks on Node Classification. ICLR 2020.

---

### Decision · Program_Chairs · 2021-09-27

**Decision:**

Accept (Poster)

**Comment:**

This paper proposes a novel bandit algorithm for sampling neighbourhoods for GNNs. The reviewers agreed that the work was of good quality, significance and originality. Some reviewers noted, and the authors are encouraged to take on board, that the current write up is quite dense and exposition could be improved by addressing this.